

# Effects of atmospheric dynamics and aerosols on the

# thermodynamic phase of cold clouds

Jiming Li[1], Qiaoyi Lv[1], Min Zhang[1], Tianhe Wang[1],

Kazuaki Kawamoto[2] and Siyu Chen[1]

5    [1]Key Laboratory for Semi-Arid Climate Change of the Ministry of Education, College

of Atmospheric Sciences, Lanzhou University, Lanzhou, China

[2]Graduate School of Fisheries Science and Environmental Studies, Nagasaki

University, Nagasaki, Japan

10    Running Head: Effects of dynamics and aerosols on the cold cloud phase

Corresponding author: Jiming Li, Key Laboratory for Semi-Arid Climate Change of
the Ministry of Education, College of Atmospheric Sciences, Lanzhou University,
Lanzhou, Gansu 730000, China. (lijiming@lzu.edu.cn)





## Abstract

Based on the 4 years (2007–2010) of data from the CloudSat 2B-CLDCLASS-LIDAR product, the European Centre for Medium-Range Weather Forecasts Auxiliary (ECMWF-AUX) product and Cloud-Aerosol Lidar and Infrared Pathfinder Satellite Observation (CALIPSO) level 2 5km aerosol layer product, this study investigates the impact of atmospheric dynamics and aerosol on cold cloud (cloud top temperature< 0ºC) phase on a global scale in order to better understand the conditions under which supercooled liquid water will gradually transform to ice phase.

Our results show that the thresholds of parameter $T_{ice}$ (is the temperature below which all clouds are ice), $T_w$ (is the temperature above which all clouds are liquid) and $n$ (is a shape parameter that controls the relationship between supercooled liquid cloud fraction (SCF) and cloud top temperature) aren't unique for the entire globe as many models adopted. The value of $T_w$ ranges from -2ºC to -6ºC at the most regions of the globe, and decreases from high latitudes to tropics. For $T_{ice}$, its value is warmer (>-26ºC) in the typical stratocumulus regions than the values at the other regions (<30ºC). The geographic distributions of parameter $n$ are closely linked to aerosol loading and meteorological parameters, and its value varies strongly from 0 to 5. By comparing the absolute and relative differences between different cloud phase schemes and observation, we suggest that the cloud phase scheme used in Community Atmosphere Model (version 3, CAM3) and CAM5 can be considered as a preferred option in the models, and the application of dynamic thresholds of $T_{ice}$, $T_w$ and $n$ will further improve the predictions of SCF, particularly over the region of poleward of 40°.

Statistical results indicate that aerosol effect on nucleation can't fully explain the all changes of cold cloud phase in our study. SCF at a given temperature also appears to be related to the different collocations of surface temperature, vertical velocity and lower-tropospheric static stability (LTSS). We find that strong vertical motion can also enhance glaciation process and reduce the SCF (or increase $n$ value) as ice nuclei





aerosol did, and force the supercooled water to glaciate at a warmer temperature. For same vertical motion, however, high LTSS (or low surface temperature) tends to increase the SCF and force the supercooled water to glaciate at a colder temperature.

Unstable atmosphere (low LTSS and high surface temperature) in those strong ascent regions favors deep convective cloud, and further exhausts the supercooled water by strong precipitation rate. Our results verify the importance and regional of dynamical factors on the changes of cold cloud phase, have potential implications for further improving the parameterization of the cloud phase and determining the climate

feedbacks.




# 1. Introduction

Clouds play an important role in regulating the Earth's radiation budget and global hydrological cycle (Stephens, 2005). However, because observations are lacking and understanding of the physical processes involved in cloud formation is insufficient, clouds are also regarded as the greatest uncertainties in climate change predictions made by various climate models (Williams et al., 2003; Zhang et al., 2005;

Klein et al., 2013). One of the primary challenges in better understanding the role of clouds in climate forcings and feedbacks involves determining how to more accurately define the cold cloud phase (cloud top temperature<0ºC) composition between 0ºC and -40ºC, with unsophisticated cloud phase schemes in GCMs (general circulation models; Li and Le Treut, 1992; Morrison et al., 2003; Tao, 2003; Tsushima

et al., 2006). Currently, many models specify the fraction of liquid-phase clouds solely as a function of temperature (Doutriaux-Boucher and Quaas, 2004; Storelvmo et al., 2008; Song et al., 2012), related ice heterogeneous nucleation processes are not considered in some models because of the poor understanding of aerosol particles' ice nucleation ability, coating conditions and nucleation modes (e.g., deposition,

immersion freezing, contact or condensation freezing) (Lohmann and Feichter, 2005). In view of the entirely different radiative and microphysical properties of ice and liquid particles, changes in the liquid-ice phase transition will significantly affect the Earth's radiation budget and precipitation efficiency (Fu et al., 1999; Fu, 2007; Sassen and Khvorostyanov, 2007; Sun et al., 2004, 2015). Thus, the oversimplification of

cloud phases in climate models inevitably leads to large biases in the study of various climate feedbacks and the sensitivity of these models.

    The Clausius-Clapeyron theory and laboratory results have indicated that liquid water particles can exist at a temperature threshold as low as -38ºC to -40ºC before homogeneous nucleation occurs (Roger and Yau, 1989). Studies based on Lidar data

and satellite observations have further verified the existence of liquid water at temperatures as low as -30ºC to -40ºC (e.g., Intrieri et al., 2002; Naud et al., 2006; Shupe et al., 2006; Morrison et al., 2011). For example, using un-polarized, ground-based Lidar data from Chilbolton in Southern England, Hogan et al. (2003)



have found that 27% of clouds between -5ºC and -10ºC in Chilbolton contain a

supercooled liquid-water layer, and this percentage falls steadily with temperature and

reaches approximately zero at temperatures below -35ºC. Giraud et al. (2001) have

used the Along-Track Scanning Radiometer (ATSR)-2 infrared data from the ERS-2

satellite to analyze the relationship between cloud phase and cloud top temperature.

Their results have indicated that the probability of ice phase clouds decreases

quasi-linearly with cloud top temperature from nearly 100% at around -33ºC to close

to 0% at -10ºC. By using polarimetric satellite data, Doutriaux-Boucher and Quaas

(2004) have also derived a global lower limit of -32°C for 100% of ice phase clouds.

However, the lowest temperature thresholds at which liquid water particles can exist

within various climate models vary dramatically from -15ºC (Smith,1990;

Doutriaux-Boucher and Quaas, 2004) to -23ºC (Weidle and Wernli, 2008) to -40ºC

(Del Genio et al., 1996; Collins et al., 2004). In addition, the relationship between the

supercooled liquid cloud fraction (SCF) and cloud top temperature (CTT) in some

models and reanalysis datasets is fixed with an exponent of 1.7 (Doutriaux-Boucher

and Quaas, 2004) or 2 (Smith, 1990; Weidle and Wernli, 2008). The unique

temperature thesholds and relations for the entire globe, regardless of their geographic

or temporal variations, eventually lead to the SCF at a given cloud top temperature

express considerable differences among GCMs. For example, the liquid water cloud

fraction at -15ºC varies from 12% to 83% in six single column models (SCMs) used

in a model comparison study of Arctic mixed-phase clouds (Klein et al., 2009;

Morrison et al., 2009). The geographic and temporal variations of SCF at a given

temperature are further complicated by several factors, including ice nuclei (IN)

concentrations and dynamic conditions such as vertical motion (Naud et al., 2006;

Choi et al., 2010; Zhang et al., 2015). Combined satellite observations and reanalysis

datasets have the potential to yield global cloud phase statistics and to clarify the

relationship between cloud phase and microphysical/dynamic processes. This

information would aid in the design and evaluation of more physically based cloud

phase partitioning schemes, improve calculations of clouds' radiative effects, and

reduce uncertainties in cloud feedbacks within GCMs.



The millimeter-wavelength cloud-profiling radar (CPR) on CloudSat (Stephens

et al., 2002) and the cloud-aerosol Lidar with orthogonal polarization (CALIOP)

(Winker et al., 2007) on CALIPSO (launched in late April 2006) can provide more

accurate data related to the vertical structure of clouds, along with cloud phase

information on a global scale (Hu et al., 2010; Li et al., 2010, 2015). The

depolarization ratio and layer-integrated backscatter intensity measurements from

CALIOP can help distinguish cloud phases (Hu et al., 2007, 2009). Using combined

CALIOP/IIR/MODIS measurements, Hu et al. (2010) have compiled global statistics

on the occurrence, liquid water content and fraction of supercooled liquid clouds; and

they have further developed a new cloud thermodynamic phase parameterization.

Cheng et al. (2012) have examined the effect of this new cloud phase

parameterization within a climate simulation by replacing the default parameterization

in the CAM4 with this new one. In addition, Choi et al. (2010) and Tan et al. (2014)

have utilized the vertically resolved observations of clouds and aerosols from

CALIPSO to analyze cold cloud phase changes and possible aerosol impacts at given

temperatures. However, systematic studies of the statistical relationship between

cloud phase and IN aerosol properties under different dynamic conditions on a global

scale have received far less attention. In this study, we combine cloud phase

information from CloudSat and CALIPSO, aerosol data from CALIPSO, and dynamic

parameters from the ECMWF-AUX and ERA-interim reanalysis datasets to

investigate the geographic and seasonal variations of different parameters' thresholds

used in the cloud phase partitioning schemes of climate models. We also perform a

preliminary evaluation of how well different cloud phase partitioning schemes can

characterize the variation of the SCF at cloud top temperatures from -40°C to 0°C; and

we further evaluate and discuss the effects of atmospheric dynamics and aerosols on

cloud phase at a given temperature.

This paper is organized as follows: a brief introduction of all datasets used in this

study is given in Section 2. Section 3 outlines the global distributions of several

important cloud phase parameters used in the models, evaluates the performance of

different cloud phase partitioning schemes and discusses the effects of atmospheric





dynamics and aerosols on a cloud's thermodynamic phase. Important conclusions and

discussion are presented in Section 4.

## 2. Datasets and methods

In the following study, 4 years (2007–2010) of data from the latest release of the

CloudSat 2B-CLDCLASS-LIDAR (version 1.0) product (e.g., radar–LiDAR cloud

classification), the ECMWF-AUX product and the CALIPSO level 2, 5 km aerosol

layer product are collected to analyze the effects of atmospheric dynamics and

aerosols on the thermodynamic phase of cold clouds on a global scale. To analyze the

regional variability of the studied parameters, we divide the globe into $2^{\circ} \times 6^{\circ}$ grid

boxes and collect a valid sample set from each grid box. Only those results and

findings derived from daytime data are provided in this study in order to support the

analysis of the radiative effects of different cloud phases in parallel studies.

### 2.1 Meteorological reanalysis dataset

In this study, the temperature profiles and surface temperatures (that is, skin

temperature) used in our analysis are taken from the ECMWF-AUX product (Partain,

2004), which is an intermediate product that contains the set of ancillary ECMWF

state variable data interpolated to each CloudSat cloud profiling radar (CPR) bin. In

addition to this information, the collocated vertical velocity parameter from the

ERA-Interim daily dataset (Dee et al., 2011) is also extracted and used in our analysis.

Here, the temperature profile is used to identify supercooled water clouds from all

water clouds, determine the aerosol and cloud layer top temperatures, and calculate

the lower-tropospheric static stability (LTSS), which is defined as the difference in

potential temperature between 700 hPa and the surface (Klein and Hartmann, 1993),

or  $\Delta\theta = T_{700}\left(\dfrac{1000}{p_{700}}\right)^{R/C_p} - T_{sfc}\left(\dfrac{1000}{p_{sfc}}\right)^{R/C_p}$ , where $p$ is pressure, $T$ is temperature, $R$ is

the gas constant of air, and $C_p$ is the specific heat capacity at a constant pressure. A

high LTSS value represents a stable atmosphere, whereas a low LTSS value

represents an unstable atmosphere. In Section 3.4, we will discuss the effects of

vertical velocity, LTSS and skin temperature on cloud phase in detail.

### 2.2 Cloud phase product





Naud et al. (2006) have indicated that cloud–radiation interactions are most sensitive to various parameters near the cloud top. Thus, we focus on the cloud top

phase (CTP) and temperature (CTT) information in this analysis. Cloud phase information is derived from the CloudSat 2B-CLDCLASS-LIDAR (version 1.0) product. Compared with the CALIPSO phase identification (lidar-only alogorithm), the 2B-CLDCLASS-LIDAR product utilizes cloud boundaries retrieved from combined CPR and CALIOP measurements, the cloud layer maximum Ze identified

with CPR, the layer integrated attenuated backscattering coefficient (IBC) from CALIOP and the temperature profile from the ECMWF-AUX product to identify three different cloud phases (ice, mixed and liquid). However, the Lidar-only phase algorithm only distinguishes the water and ice phases of a cloud by using the Lidar depolarization ratio and layer integrated attenuated backscattering coefficient (IBC)

(Hu et al., 2007, 2009). Due to the strong multiple scatter effect in the Lidar depolarization measurements, as well as Lidar's limited ability to penetrate optically thick clouds, CALIPSO's Lidar-only algorithm is restricted in its ability to identify mixed-phase clouds; in particular, it is unable to penetrate the supercooled liquid layer to detect the ice layer (Zhang et al., 2010), and it is unable to distinguish pure liquid

clouds from mixed-phase clouds. Nevertheless, only cloud top information is needed in this study. Therefore, the differences between these two algorithms should not result in abrupt or obvious changes in cloud phase fractions. Given the importance of multilayered cloud systems (Huang et al., 2005, 2006a; Lv et al., 2015), we obtain the cloud phase information of every cloud layer in each sample profile and further group

every cloud layer into separate temperature bins (1ºC interval) according to the ECMWF-AUX temperature profiles and 2B-CLDCLASS-LIDAR cloud layer top heights. For mixed-phase clouds, we define the cloud as ice-topped or liquid-topped based on the "water_layer_top" information. If the temperature of the water layer in a mixed-phase cloud is equal to or lower than the mixed-phase cloud top temperature, it

is classified as a liquid-topped cloud. Otherwise, it is classified as ice-topped. Furthermore, liquid phase clouds are divided into warm water-phase (CTT≥0) and supercooled water-phase (CTT<0) according to their cloud top temperature. Only



those supercooled water phase clouds with a CTT between -40ºC and 0º C are further analyzed in this study. Here, we define the supercooled water cloud fraction (SCF) in

a given temperature bin as the ratio of the number of liquid phase samples and the total (liquid+ice) samples gathered in a $2^{°} \times 6^{°}$ grid box.

### 2.3 Aerosol types and relative frequency

Aerosol data are obtained from the CALIPSO level 2, 5 km aerosol layer product. Using scene classification algorithms (SCA), CALIPSO first classifies the

atmospheric feature layer as either a cloud or aerosol by using the mean attenuated backscatter coefficients at 532/1064 nm, along with the color ratio (Liu et al., 2009). A confidence level for each feature layer is also reported in the level 2 products. Using the surface type, lidar depolarization ratio, integrated attenuated backscattering coefficient and layer elevation, aerosols are further distinguished as desert dust,

smoke, polluted dust, clean continental aerosol, polluted continental aerosol, and marine aerosol (Omar et al., 2009). Mielonen et al. (2009) have used a series of Sun Photometers from the Aerosol Robotic Network (AERONET) to compare CALIOP and AERONET aerosol types and have found that 70% of the aerosol types from these two datasets are similar, with the closest similarities occurring in dust and polluted

dust types. Mineral dust from arid regions has been widely recognized as an important source of ice nuclei in mixed-phase clouds because of its nucleation efficiency and abundance in the atmosphere (Richardson et al., 2007; DeMott et al., 2010; Atkinson et al., 2013). In addition to dust, some studies have also verified the potential ice nucleation ability of polluted dust and smoke at cold temperatures (Niedermeier et al.,

2011; Cziczo et al., 2013; Zhang et al., 2015). For example, by using satellite lidar observations, Tan et al (2014) have found negative temporal and spatial correlations between the supercooled liquid cloud fraction and the polluted dust and smoke aerosol frequencies at the −10ºC, −15ºC, −20ºC, and −25ºC isotherms, although those correlations are weaker than those found between dust frequencies and the

supercooled liquid cloud fraction. As a result, we combine the dust, polluted dust and smoke information from CALIPSO to further analyze the relationship between aerosols and the SCF in this study. Given the difficulty of quantifying the



concentration of IN aerosols (here, IN aerosols are the sum of dust, polluted dust and smoke), this study utilizes the relative occurrence frequency of IN aerosols to quantify this variable instead. We first group every IN aerosol sample from each observation profile into a different temperature bin (1ºC interval) according to the ECMWF-AUX temperature profiles and CALIPSO aerosol layer top height measurements. Then, following Choi et al. (2010), we define the frequency of IN aerosols within a given temperature bin as the ratio of the number of IN aerosol samples to the total number of observation profiles in the same temperature bin and grid. Finally, we calculate the relative occurrence frequency of IN aerosols with respect to the highest IN aerosol frequency. The relative occurrence frequency of aerosols is indicative of the temporal and spatial variability of IN aerosols compared to the maximum occurrence frequency (Choi et al., 2010). We remove those aerosol layers with low confidence values (that is, those with an absolute value lower than 50) from the dataset (approximately 6.5% of all aerosol layers).

## 3. Results

### 3.1 Cloud phase partitioning schemes in GCMs

Recently, several ice nucleation processes based on theoretical and empirical studies have been developed to more explicitly represent these processes in certain climate models. These new schemes have indicated that the liquid cloud fraction should depend not only on temperature but also on the presence of aerosols that have undergone ice nucleation (Storelvmo et al., 2008; Gettelman et al., 2012). However, many models still specify the liquid-phase cloud fraction solely as a function of temperature (Doutriaux-Boucher and Quaas, 2004; Hu et al., 2010; Song et al., 2012) because the microphysical and dynamic processes of cloud formation are not yet fully understood. For example, Choi et al. (2014) have summarized the cloud phase partitioning schemes used in various climate models and have studied the influence of cloud phase composition on climate feedbacks. They outlined the two cloud phase schemes that are widely used in present models. Scheme 1 can be written as:





$$f = \left(\frac{T - T_{ice}}{T_w - T}\right)^n \qquad (1)$$

For scheme 2, the liquid fraction $f$ can be expressed as:

$$f = \exp[-(\frac{T_w - T}{15})^n] \qquad (2)$$

where $T$ is temperature, $T_{ice} \leq T \leq T_w$, $T_w$ is the temperature above which all clouds

are liquid, $T_{ice}$ is the temperature below which all clouds are ice, and $n$ is a shape

parameter that controls the slope of $f(T)$ between -40º and 0º. Based on table 1 of Choi

et al. (2014), we select several models and list the values of these parameters in Table

1. Obviously different thresholds of these parameters and different cloud phase

schemes in climate models indicate that an inability for models to accurately separate

the cloud phases, and the large biases and inconsistency between the models may be

because these thresholds are based on aircraft observations or field experiments from

different regions. Thus, a constant global threshold would probably introduce large

uncertainty in the simulation of cloud feedbacks, and the spatial and temporal

variations of these parameters should be considered in the future parameterization of

cloud phase partitioning.

### 3.2 Global distributions of $T_w$, $T_{ice}$ and $n$

Based on the processes outlined in Section 2.2, the SCF for each temperature bin

(1K) of every grid can be derived. The value of $T_w$ in each grid equals the temperature

above which all SCFs equal 1, whereas the value of $T_{ice}$ in each grid equals the

temperature below which all SCFs equal 0. After obtaining values for $T_{ice}$ and $T_w$, the

value of $n$ can be further determined by performing nonlinear fitting to $T_{ice}$, $T_w$, $f$ and

$T$ using Eq.(1). In the following analysis, we determine that scheme 1 (Eq. (1)) better

simulates the variation of SCF with temperature than scheme 2 (Eq. (2)), and thus

only the distributions of $n$ for scheme 1 are provided in this section.

Fig. 1 shows the geographic and seasonal variations of $T_w$ values across the $2^{\circ} \times 6^{\circ}$

grid boxes based on the 2B-CLDCLASS-LIDAR product. The gaps (no color)

indicate missing data or areas where the supercooled water cloud fraction doesn't

reach 1 between -40 ℃ and 0 ℃. Those grids in which $T_w$ is higher than 0 ℃ are





excluded in this study. These grids almost all located within typical subsidence

regions (e.g., stratocumulus regions) where strong subsidence favors low cloud

formation and suppresses ice or mixed-phase cloud generation (Yuan and Oreopoulos,

2013). Fig. 1 clearly illustrates that the value of $T_w$ ranges from -2ºC to -6ºC across

the majority of the globe; moreover, no clear seasonal variations are found in our

results for $T_w$. At high latitudes, $T_w$ ranges from -2ºC to -3ºC; this value decreases

from the high latitudes to the tropics. Our analysis indicates that the current $T_w$ value

used in CAM3 may be too low and may result in the overestimation of supercooled

water clouds at lower altitudes (see Fig. 7), whereas the $T_w$ value used in CAM5 is

consistent with the distribution of $T_w$ for most regions across the globe. For the

ERA40 reanalysis dataset (Weidle and Wernli, 2008) and other models such as the

LMDZ (Doutriaux-Boucher and Quaas, 2004), a high threshold is generally adopted,

which probably results in the underestimation of supercooled water clouds at lower

altitudes. Fig. 2 illustrates the geographic and seasonal variations of $T_{ice}$ at the $2^{°} \times 6^{°}$

grid box scale. The warmest $T_{ice}$ values for each season generally occurs in typical

stratocumulus regions and in northern Africa; these values are warmer than -26ºC,

indicating that supercooled water clouds in these regions are restricted to warmer

atmospheric levels than are found throughout the majority of the world, where $T_{ice}$

values are almost always below -30ºC. Although the differences between these

observations and the models are clear, the thresholds used in the CAM3 (or 5) cloud

phase scheme and GISS are relatively reasonable compared with those values used in

the ERA40 reanalysis dataset (Weidle and Wernli, 2008) and other models. Generally,

$T_{ice}$ values can reach lower temperatures in a clean or IN-poor environment. However,

our results show that the geographic and seasonal variations in $T_{ice}$ are also negligible

under different aerosol loading conditions, indicating that the combination of several

factors, such as IN, vertical motion or other dynamic parameters affect the distribution

of $T_{ice}$.

In each model's cloud phase scheme, the shape parameter $n$ controls the slope of

the curve between temperature and the supercooled water cloud fraction. For example,

in scheme 1, a large $n$ value corresponds to a low liquid cloud fraction at a given




cloud top temperature $T$, $T_{ice}$, and $T_w$. This relationship is showed in Fig. 3, which

depicts the apparent decreases of grid-mean SCF from 0.6 to 0.3 with corresponding

increases in parameter $n$ from 0.5 to 5.5. Here, the grid-mean SCF is the averaged

value of the supercooled water cloud fraction from all cloud top temperature bins

ranging from -40ºC to 0ºC within a given grid cell, and the color bar represents the

number of grid cells within a 4 year period. Further, Fig. 4 shows the clear geographic

and seasonal variations of parameter $n$. Based on Fig. 4, we find that $n$ is

approximately 1 at 60º poleward and varies strongly from 0 to 5 throughout a majority

of the globe. Larger values (equal to or greater than 3) locate at the mid-latitudes of

the northern hemisphere, South America and the mid-latitude oceans of the southern

hemisphere. Especially at the mid-latitudes of the northern hemisphere, the value

reaches even 4 or 5. Given the values of $n$ used in the CAM 3 (Collins et al., 2004)

and CAM 5 (Song et al., 2012), it is clear that the CAM 3 and CAM5 better simulate

the relationship between temperature and SCF at the high-latitudes (60º poleward);

meanwhile, the values of $n$ adopted in the ERA40 (Weidle and Wernli, 2008) and

LMDZ (modified version) (Doutriaux-Boucher and Quaas, 2004) are consistent with

observed results in only some regions during certain seasons (e.g., the Pacific Ocean

during all seasons except summer, and the high-latitudes of the northern hemisphere

during summer and Russia year-round). For other regions such as Asia, South

America and the North Pacific, large $n$ values in these models definitively indicate the

models' inability to accurately simulate the relationship between temperature and SCF

locally.

### 3.3 Evaluation of cloud phase partitioning schemes

Following the process used to evaluate scheme 1, we also derive the parameter $n$

for scheme 2 (not shown). After inputting the dynamic thresholds of $T_{ice}$, $T_w$ and $n$ into

Eq. (1) and Eq. (2), we are able to calculate the SCF of each cloud top temperature bin

within each geographic grid for the different schemes, and we further evaluate which

scheme is better able to simulate the variation of SCF with temperature in each grid.

Here, we define the grid mean absolute value of the difference between calculated and

observed SCF (absolute difference) at each temperature bin as follows:



$$Abs\_dif = \sum_{T=-40^oC}^{0^oC} |SCF^T_{calculated} - SCF^T_{observed}| \Big/ 41 \qquad (3)$$

Following the same logic, the grid mean relative difference can be written as:

$$\mathrm{Re}\_dif = \sum_{T=-40^oC}^{0^oC} (SCF^T_{calculated} - SCF^T_{observed}) \Big/ 41 \qquad (4)$$

where $T$ is the cloud top temperature, and $SCF^T_{observed}$ and $SCF^T_{calculated}$ are the observed

and calculated SCFs from scheme 1 and 2, which are determined by inputting the

dynamic (or fixed) thresholds of $T_{ice}$, $T_w$ and $n$ into Eq. (1) and Eq.(2), respectively. In

addition, 41 is the number of cloud top temperature bins from -40ºC to 0ºC. Figs. 5

and 6 compare the geographic distributions of absolute and relative differences

(annual means) for different schemes, respectively. Compared with scheme 2, which

is used in the GISS model (see Fig. 5b), the cloud phase partitioning scheme (scheme

1) is used in CAM 3 (or 5) better simulate the variation of SCF with temperature

almost everywhere, especially at the mid- and high-latitudes (see Fig. 5a). For

example, the absolute difference for scheme 1 (Fig. 5a) is smaller than 0.08 at 40º

poleward, with a large value (0.12) only apparent in the oceanic regions of the

subtropics. However, in scheme 2 (Fig. 5b), the differences across most regions of

globe still exceed 0.16, even when dynamic thresholds are inputted. Figs. 5c and 5d

further illustrate the absolute difference between CAM3 (and 5) calculated SCFs and

observed SCFs. At present, CAM 3 and CAM 5 still rely on unique temperature

thresholds and the $n$ value identified in scheme 1 for the entire globe, which has led to

considerable variations in absolute difference values compared with those shown in

Fig. 5a. By comparing Figs. 5b and 5d, we find that the distributions and magnitudes

of the absolute differences are very similar, and even the high-latitude values from

CAM5 are smaller than those results derived from scheme 2. These figures further

verify the importance of the cloud phase partitioning scheme in general circulation

models. Although the schemes used in CAM 3 and 5 are similar, the difference is

more apparent in CAM 3 than CAM 5 (see Fig. 5c). In fact, based on Eq.(1) and Table

1, it is clear that the difference between Fig. 5c and 5d is mainly caused by the





unreasonable threshold of $T_{ice}$ in the CAM3, which yields an additional 17% difference in values compared to the CAM5. However, the limits of CAM 5 are still apparent in the northern hemisphere, as seen by comparison of Figs. 5a and 5d, especially at mid-latitudes where the effect of IN aerosols is important (the difference

can reach 0.2). Thus, scheme 1's ability to consider the dynamic thresholds of $T_{ice}$, $T_w$ and $n$ may further improve the prediction of the supercooled water cloud fraction at different temperatures, particularly poleward of 40°, thus making it a preferred option in general circulation models. For relative difference values (Fig. 6), a clear underestimation (exceeding -6%) occurs poleward of 60°S over the ocean (Figs. 6b

and 6d), whereas SCF is overestimated significantly (14%) in other regions by the models, especially CAM 3 (Fig. 6c). However, the relative difference for scheme 1, which consider the dynamic thresholds of $T_{ice}$, $T_w$ and $n$, ranges from only -0.04 to 0.02 (Fig. 6a).

For studying the vertical distribution of zonal mean SCF with temperature, and

further evaluating the absolute and relative differences between calculated and observed SCFs for scheme 1 are mainly from which temperature bin, the Fig. 7 give us some new insights. For example, Fig.7a depicts the vertical distribution of zonal mean SCFs (annual mean) with temperature based on the observation. Here, the SCF at each temperature bin of each latitude belt is the averaged value of SCFs of all grids

at this temperature in this latitude belt. Fig. 7a further illustrates that nearly all of the SCF values are close to 1 or 0 at temperatures above -5ºC and below -35ºC, respectively. Thus, the $T_{ice}$ and $T_w$ temperature thresholds used in CAM5 are probably more reasonable values than those in CAM3, at least at the zonal mean level. In addition, the meridional asymmetry (or hemispheric asymmetry) of the SCF values is

apparent, with southern hemispheric values exceeding those in the northern hemisphere. This clear difference is particularly prominent at the temperature range from -15ºC to -30ºC poleward of 50°S, where the difference between SCFs in the southern and northern hemispheres sometimes exceeds 20% and reaches 40% at -25ºC in the polar region (figure not shown). Given the similar vertical distributions of the

SCF values, Fig. 7b provides a typical example of the SCF profile differences





between the dynamic thresholds derived by scheme 1 and the observed values. Overall, the difference is small in most temperature bins (with bias ranging from -9% to 9%) and is primarily concentrated in the southern hemisphere. A clear underestimation (or overestimation) exceeding 12% is produced by scheme 1 for the

temperature range -20ºC to -25ºC (or about -10ºC) at 40°S to 40°N, and especially approximately 40°S; however, CAM3 generally produces a substantial overestimation almost everywhere except for the temperature range -20ºC to -25ºC poleward of 60°S, where a slight underestimation occurs (from 9% to18%). The significant overestimation of SCF by CAM3 is particularly prominent in the lower atmosphere at

temperatures between -10ºC to -20ºC for almost every latitude belt (where the bias reaches up to 45%). For those regions where temperatures drops below -20ºC, the SCF bias decreases to approximately 30%; nevertheless, SCF values are still significantly overestimated. Compared with CAM3, the bias is smaller in CAM5 (Fig. 7d), although the patterns of differences are very similar. At the temperature range

-20ºC to -25ºC poleward of 60°S, the underestimation of values (approximately 30%) is more apparent than those results for the same range derived from CAM3. Fig. 6 allows us to infer that the clear bias in relative differences poleward of 60°S produced by CAM5 is primarily caused by the underestimation of SCFs in the -20ºC to -25ºC temperature range, whereas the apparent overestimation of SCF from -10ºC to -20ºC

contributes most of the bias at other latitude belts.

### 3.4 Effects of atmospheric dynamics and aerosols on cloud thermodynamic phase

The above analysis demonstrate the inability of current models to accurately simulate the vertical and geographic variations of the supercooled cloud fraction, due to incomplete knowledge of the underlying physical processes related to cloud phases.

In fact, several factors in addition to cloud top temperature, including IN and vertical motion, probably together affect the distribution of the supercooled cloud fraction, especially the relationship between temperature and supercooled water clouds. For example, the distribution of parameter $n$ at the mid-latitudes of the northern hemisphere may be largely related to dust aerosols that can serve as IN and thus

enhance the glaciation occurrences at lower relative SCFs than in other regions of the



globe at a given cloud top temperature, e.g., -20ºC (Fig. 8). For example, as showed in Fig. 8, the SCF at -20ºC is only approximately 0.12 (especially at spring) at the mid-latitudes of the northern hemisphere, whereas SCF reaches 0.7 at the high-latitudes of the southern hemisphere (that is, poleward of 60°S). Our results are

consistent with those from previous studies from Choi et al (2010) and Tan et al (2014), which have verified that the regional differences in the supercooled water cloud fraction at -20ºC are highly correlated with the dust frequency above the freezing level. However, by collocating this variable with the geographic and seasonal distributions of relative aerosol occurrence frequency (RAOF) at -20ºC (see Fig. 9),

we find that the SCF still has a low value at the mid-latitudes of the northern hemisphere during the summer season, even though the IN aerosol loading is insignificant at -20ºC in these regions. As in the findings of Choi et al (2010), our results also show that the persistent low SCF (or large $n$) throughout the year at −20ºC (see Fig. 8) could not be explicitly related to IN aerosol frequency because the RAOF

is significantly lower in central South America. These results indicate that aerosols' effect on nucleation cannot fully explain all changes of cold cloud phase in our study; In other words, there is no evidence to suggest that its effect is always dominant at each altitude of each region.

   Besides aerosol effect, what is the role of meteorological effect in determining

cloud phase change, especially at those regions which aerosol effect on nucleation isn't a first-order influence due to low IN aerosol frequency To further discuss this question, we analyze the seasonal and zonal variations of SCF and RAOF at -20ºC, LTSS and 500 hPa vertical velocity (see Fig. 10). For the tropical region (from 20°S to 20°N), the RAOF is very low (approximately 0.005) and the seasonal variation is

negligible. However, the corresponding SCF undergo a clear seasonal change. For example, opposing SCF distributions are found during the summer and winter, and the maximum difference reached 15%. Opposite distributions are primarily linked to distinctly different atmospheric vertical motions, whereas surface temperature (figure not show) and LTSS contribute minimal amounts, due to their weak seasonal variation

above this region. During the summer, the inter-tropical convergence zone (ITCZ) is





correlated with deep convective clouds shift to the northern hemisphere and has strong vertical velocity. In contrast, the strength of atmospheric vertical motion in the northern hemisphere tropics decreases during the winter season, due to the ITCZ shift to the southern hemisphere. Poleward of 40°N, the inconsistency of seasonal

variations in SCF and RAOF is particularly apparent. For example, maximum RAOF and SCF are visible during winter at the middle and high latitudes of the northern hemisphere, whereas these values both decrease during the summer. Another apparent phenomenon is that winter RAOF values are larger than spring values poleward of 40°N. We recalculate the RAOF by considering only IN dust aerosols, and find that

this phenomenon still exists. These results suggest that the trend is real and is not fully caused by combined polluted dust and smoke frequencies, thus verifying the importance of meteorological effects on cloud phase changes. Figs. 10a and 10b show how the differences between winter and summer SCF may be linked to the seasonal variations in surface temperature and atmospheric stability (LTSS), whereas seasonal

changes in vertical motion are weak and probably minimally affect SCF at the mid- and high-latitudes of the northern hemisphere. The results also show that the effects of different meteorological factors on cold cloud phase have regional characteristics. That is, low surface temperature and high LTSS (or a stable atmosphere) inhibit ice nucleation and push supercooled water to colder temperatures at mid- and

high-latitudes, whereas strong vertical motion enhances ice nucleation in the tropics.

To further quantify the effects of aerosol and meteorological factors on cold cloud phase, we group the RAOFs of grids into several RAOF bins within each specified vertical velocity, surface temperature or LTSS bin in order to analyze the relationship between the studied parameters (SCF at -20°C, $T_{50}$ and $n$) and aerosol loading under

different meteorological conditions. Here, $T_{50}$ is defined as the cloud top temperature for exactly 50% of supercooled water clouds (Naud et al. 2006) and can be derived by inputting the $T_{ice}$, $T_w$, $n$ and $f$ (50%) thresholds into Eq. (1). Fig. 11 gives the seasonal and geographic variations of this parameter. Based on Fig. 11, we find that $T_{50}$ in the high-latitude regions is lower (around -20°C) than that in other regions, especially the

middle latitudes of the two hemispheres, where $T_{50}$ reaches -10°C. The apparent



difference (approximately 10K) in $T_{50}$ values between the high and middle latitudes reflects the general tendency for supercooled water clouds to persist at colder temperatures in the high latitude regions, and this result based on global observation also further supports the findings of Naud et al. (2006), who have analyzed MODIS

data collected over the North Atlantic and Pacific Ocean basins. Naud et al. (2006) have also found that the warmest $T_{50}$ values in each studied subregion generally occur in areas of ascent and heavy precipitation. However, by analyzing the distributions of $T_{50}$ on a global scale, our results indicate that strong subsidence areas also tend to generate warm $T_{50}$ values. This phenomenon is particularly apparent in Fig. 12, which

depicts the dependences of $T_{50}$, $n$ and SCF on the RAOF and 500 hPa vertical velocity. Notably, the temperature bins used to calculate the RAOF differ for $T_{50}$, $n$ and SCF, based on their global distributions (See Fig. 4, Fig. 8 and Fig. 11). For example, parameter $n$ reflects the relationship between SCF and CTT at the temperature ranges from -40ºC to 0ºC; thus, the calculation of RAOF for $n$ considers aerosols at all

temperature bins from -40ºC to 0ºC. However, only the aerosol samples at ±2 bins around -20ºC are used to calculate the RAOF for SCF at -20ºC. For $T_{50}$, the calculation of RAOF is primarily based on aerosol samples from the -20ºC to 0ºC temperature bins. We separate the relationship between $T_{50}$ and RAOF into three groups based on the strength of the 500 hPa vertical velocity (e.g., 0<|vertical

velocity|<=25 hPa/day, 25<|vertical velocity|<=50 hPa/day and |vertical velocity|>50 hPa/day). Such grouping ensures a sufficient number of samples available in each bin (at least hundreds of samples in each bin) to ensure statistical significance. The error bars correspond to the ±5 standard error. Here, the standard error (SE) is computed as:

$SE = SD / \sqrt{N}$ , where SD is the standard deviation of the data falling in an RAOF

bin and vertical velocity bin, and $N$ is the sample number in each bin. As RAOF increases (Fig. 12a), $T_{50}$ gradually tends to increase from around -20ºC (IN aerosols absent) to approximately -12ºC (high aerosol loading). The continuously increasing trend verifies that the existence of IN aerosols can hasten the glaciation of supercooled droplets through the Bergeron-Findeisen growth mechanism of ice





crystals (e.g., Pruppacher and Klett, 1978). The value of $T_{50}$ tends to increase when the 500 hPa vertical velocity increases from <25 hPa/day to >50 hPa/day. That is, strong vertical velocity can also force supercooled water to glaciate at a warmer temperature. On average, the maximum difference in $T_{50}$ with and without aerosol loading reaches 8K, whereas the effects of different vertical velocities result in a

difference of approximately 2K within the same RAOF bin. This result indicates that the effect of vertical motion on $T_{50}$ is relatively smaller than the effect of aerosols. Based on the MODIS retrieval data, Naud et al. (2006) have analyzed the frontal ascent region of storm systems. They have found that glaciation occurs on average at the warmest temperature (warm $T_{50}$), where the strongest precipitation rates and

updrafts occur. In addition, a vigorous ascent may maintain conditions near water saturation, and $T_{50}$ values close to those at which the Bergeron-Findeisen process operates most efficiently may indicate that the process limits the existence of liquid droplets at colder temperatures.

       Fig. 12b illustrates the relationships between parameter $n$ and aerosol loading for

different vertical velocities. As with Fig. 12a, the $n$ value gradually increases as RAOF increases, and those regions with a large vertical velocity have large $n$ values. On average, the maximum difference in $n$ values for areas with and without IN aerosol loading is approximately 2, and the $n$ bias caused by vertical motion is relatively less than (approximately 0.5) than the aerosol effect. Large $n$ values

correspond to small SCF values, and Fig. 12c clearly shows the decreasing tendency of SCF at -20ºC as the RAOF and 500 hPa vertical velocity increase. The logarithmic intervals of RAOF (see the X-axis of Fig. 12c) indicate that there is a semi-logarithmic relationship between SCF and RAOF (Choi et al., 2010). For relatively clean air without IN aerosols, the SCF at a cloud top temperature of -20 ºC

exceeds 40% when the 500 hPa vertical velocity is smaller than 25 hPa/day. This value gradually decreases to 15% under high aerosol loading conditions when the 500 hPa velocity is also high (>50 hPa/day). The clearly decreasing trend in SCF with increasing RAOF is concordant with previous studies' conclusions based on CALIPSO measurements (Choi et al., 2010; Tan et al., 2014). These distinctly




separate curves verify that the vertical velocity indeed significantly affects ice nucleation even in the case of high aerosol loading. In fact, the effect is particularly important in those regions without IN aerosols where the SCF bias can reach 15%, such as in the tropics. For the same RAOF bin, a large vertical velocity can enhance the glaciation process and reduce the SCF, which partly explains why the persistent

low SCF at -20ºC across central South America throughout the year could not be explicitly related to aerosol frequency. Overall, different vertical motions lead to 10% mean SCF differences, and the bias is comparable to the effect of aerosols on cloud phase changes when the vertical velocity is limited to the same speed.

Similarly, Figs. 13 and 14 show how $T_{50}$, $n$ and SCF depend on aerosols, surface

temperature and atmospheric stability, respectively. Here, surface temperature is classified into three levels: high, where the surface temperature$\geq$285 K; medium, where 270 K$\leq$surface temperature<285 K; and low, where surface temperature<270K. Generally, high surface temperatures enhance ice nucleation and reduce SCF values (or have large $n$ values). As with strong vertical velocity, warm surface temperatures

can also force supercooled water to glaciate at warmer temperatures. The difference in $T_{50}$ values at different surface temperatures reaches 10K without aerosol loading and gradually decreases to approximately 3K with high aerosol loading. On average, the different surface temperatures lead to 25% SCF differences, and the bias is larger than the difference caused by different vertical motions (10%). This is also comparable to

the effect of aerosols on cloud phase changes when the vertical velocity is limited to the same level. High LTSS represents a stable atmosphere, whereas low LTSS represents an unstable atmosphere. In Fig. 14, a stable atmosphere (LTSS>=19K) can be seen to inhibit ice nucleation and enhance the SCF, and it is associated with colder $T_{50}$ values than an unstable atmosphere (LTSS<=14K). Although the effects of LTSS

on these parameters are not as apparent as those of surface temperature and vertical motion, some interesting results still are captured. Naud et al. (2006) have found that outside the frontal ascent zone, $T_{50}$ is not uniformly warm everywhere the mean strength of the 500 hPa vertical velocity is high. They have suggested that vigorous updrafts either suppress ice formation or advect supercooled water to the colder cloud





top level because vigorous updrafts do not leave enough time for supercooled droplets

to glaciate as ice crystals (Bower et al., 1996). In fact, these two opposite mechanisms

may correspond to storms of different intensities, different cloud systems within

different atmospheric stability levels (convective cloud or stratiform frontal cloud) or

different locations with the cloud (cloud top or inside the cloud) (Bower et al., 1996;

Naud et al., 2006). In our results, we find that Naud's results outside the frontal ascent

zone may be interpreted using the LTSS and surface temperature. For the same

vertical motion, high LTSS tends to reduce the $T_{50}$ to a cold temperature. In addition,

stratiform clouds can be generated more easily within a stable atmosphere; thus,

different LTSS values are linked to different cloud systems. Compared with a

convective cloud system, which requires a warmer surface temperature and lower

LTSS, stratiform clouds have a weak precipitation rate and inhibit the exhaustion of

supercooled water. Thus, different $T_{50}$ values within and beyond the frontal ascent

zone for similar vertical motions in Naud et al. (2006) actually reflect different

surface temperatures and LTSS. By performing a similar analysis at different latitudes,

we further find that the effect of LTSS on cloud phase is obvious at middle and high

latitudes, particularly in the northern hemisphere, where shallow stratiform clouds

such as altostratus, stratus and nimbostratus clouds are frequent (Wang and Sassen,

2001; Sassen and Wang, 2008; Li et al., 2015). Due to the wide distribution of land at

the middle and high latitudes of the northern hemisphere, seasonal variations in

surface temperature result in significant differences in LTSS during different seasons,

which further causes a difference in cloud types and amounts over this region. By

comparing the different cloud types and covers at different seasons using the

2B-CLDCLASS-LIDAR, we find that shallow stratiform cloud covers such as,

altostratus, stratus and nimbostratus clouds indeed are greater during winter than

summer, providing a reason for why the SCF and RAOF are both larger during the

winter than the summer.

In the cloud phase partitioning schemes of models, we primarily focus on the

effects of aerosols and dynamic factors on the parameter $n$. However, we also perform

a similar analysis for $T_w$ with $n$, and we find that the value of $T_w$ systematically





increases with increases in IN aerosols from -9ºC to -4ºC, indicating that high aerosol

loading can enhance the temperature of glaciation. However, the effect of dynamic

factors on $T_w$ is negligible. In addition, the stable relationship between aerosols (or

dynamics) and $T_{ice}$ is not evident in our results, indicating the complexity of the

distribution of $T_{ice}$. Based on the above analysis, the current model (e.g., CAM5)

already provides relatively reasonable $T_{ice}$ and $T_w$ values compared with other models.

Its bias toward SCF is primarily caused by the unreasonable presentation of parameter

$n$, which is closely linked to aerosol loading and meteorological factors. These results

thus suggest that the effects of dynamics and aerosols on the parameters (especially

for parameter $n$) in the studied cloud phase partitioning schemes are very important

and should be considered in the parameterization of cloud phase in future studies in

order to further improve the calculation of cloud radiative effects related to cloud

phase changes.

### 4. Conclusions and Discussion

        Changes in cloud phase can significantly affect the Earth's radiation budget and

global hydrological cycle (Sassen and Khvorostyanov, 2007; Choi et al., 2010). Based

on 4 years (2007–2010) of cloud phase data and aerosol products from CloudSat and

CALIPSO, as well as meteorological parameters from the ECMWF-AUX and

ERA-Interim products, this study investigate the effects of atmospheric dynamics and

aerosols on the thermodynamic phase of cold clouds on a global scale. Although some

statistical results reasonably agree with previous research, new insights are also

achieved in this paper.

        For certain current GCMs, unique temperature thresholds (for $T_{ice}$ and $T_w$) and

relationships (for parameter $n$) are used in the models' cloud phase partitioning

schemes, regardless of their geographic or temporal variations, which may result in

considerable differences regarding the estimation of the supercooled liquid cloud

fraction at a given cloud top temperature between the GCMs. By using the

observations from space lidar and radar, we find that the value of $T_w$ ranges from -2ºC

to -6ºC across most regions of the globe; moreover, there is no clear seasonal variation

in our results for $T_w$. At the high-latitudes, $T_w$ ranges from -2ºC to -3ºC; this value



decreases from high latitudes to the tropics. $T_{ice}$, is warmer (>-26ºC) in typical

stratocumulus regions and northern Africa when compared to values across the rest of

the world, where $T_{ice}$ is almost exclusively below -30ºC. These results verify the

reasonableness of thresholds in CAM5 compared to other models, which may

overestimate (or underestimate) supercooled water clouds at lower atmospheric levels

on the basis of the $T_w$ threshold used. The geographic and seasonal distributions of

parameter $n$ are closely linked to aerosol loading and meteorological parameters (e.g.,

vertical velocity, LTSS and surface temperature). Our results indicate that the value of

$n$ varies strongly from 0 to 5 across the majority of the globe. High values (equal to or

greater than 3) occur at the mid-latitudes of the northern hemisphere, South America

and the mid-latitude oceans of the southern hemisphere. Values of n in the CAM3 and

CAM5 models best illustrate the relationship between temperature and SCF at the

high-latitudes (60º poleward). By comparing the absolute and relative differences

between different cloud phase schemes and remote-sensing observations, we suggest

that scheme 1 used in CAM3 and CAM5 is a preferred option in the models, and the

application of dynamic $T_{ice}$, $T_w$ and $n$ thresholds should further improve the

predictions of the supercooled water cloud fraction for different temperatures,

particularly over the region poleward of 40°.

To clarify the roles of meteorological factors and aerosol loading in determining

cloud phase changes and further provide observational evidence for the design and

evaluation of a more physically based cloud phase partitioning scheme, we perform a

series of analyses that investigate the effects of atmospheric dynamics and aerosols on

the thermodynamic phase of clouds on a global scale. Statistical results indicate that

aerosols' effect on nucleation can't fully explain all cold cloud phase changes,

especially in those regions where aerosols' effect on nucleation is not a first-order

influence (e.g., due to low IN aerosol frequency). As with the effects of IN aerosols,

we find that strong vertical motion enhances the glaciation process, reduces the SCF

(or increases the $n$ value), and forces the supercooled water to glaciate at a warmer

temperature. For the same vertical motion, however, high LTSS (or low surface

temperature) tends to increase the SCF and force the supercooled water to glaciate at a



colder temperature. These two opposite mechanisms may correspond to different cloud systems (e.g., convective clouds or stratiform frontal clouds) or to different precipitation intensities. An unstable atmosphere (low LTSS and high surface temperature) in those strong ascent regions favors the formation of deep convective clouds and exhausts the supply of supercooled water through a strong precipitation

rate. A stable atmosphere (high LTSS and low surface temperature) favors the formation of shallow stratiform clouds and can inhibit the exhaustion the supercooled water via a weak precipitation rate. These results are consistent with partial findings from previous studies (Naud et al., 2006; Choi et al., 2010) and may help in interpreting some confusing phenomena observed in previous and our studies (Choi et

al., 2010). For example, these results explain why the values of SCF and RAOF during the winter are both larger than values obtained during the summer at the middle and high latitudes of the northern hemisphere.

Previous studies have mainly focused on warm water cloud systems (Li et al., 2011, 2013; Kawamoto and Suzuki, 2012, 2013) or dust properties retrieval and

simulations (Huang et al., 2010; Bi et al., 2011; Liu et al., 2011; Chen et al., 2013) or have demonstrated the importance of dust on cloud properties (Huang et al., 2006b, 2006c, 2014; Su et al., 2008; Wang et al., 2010). However, systematic studies on the statistical relationship between cold cloud phase (in particular, supercooled water clouds) and IN aerosol properties under different dynamic conditions on a global scale

have received far less attention. Our results, which are based on global observations, verify the effects of dynamic factors on cloud phase changes and illustrate that these effects are regional, thus suggesting potential implications for further improving the parameterization of cloud phases and determining the climate feedbacks. General circulation model (GCM)-simulated storm tracks move poleward (Yin, 2005), as do

the associated water clouds. The difference in albedo feedback among different models is primarily a result of the differences in the poleward redistribution of cloud-based liquid water and is related to differences in mixed-phase cloud algorithms (Tsushima at al., 2006): those models that produce more supercooled water clouds have a higher sensitivity. However, with global warming, a number of studies have





shown that spring dust storm frequencies negatively correlate with local surface air temperatures and have shown a downward trend over the past 50 years (Qian et al., 2002; Zhu et al., 2008). In addition, the warming of surface temperatures in recent decades has been enhanced relative to mean global warming by approximately 50% in the United States, a factor of 2–3 in Eurasia, and a factor of 3–4 in the Arctic and the

Antarctic Peninsula (Hansen et al., 2010). It is uncertain how these trends will affect cloud phase changes and whether more ice or more supercooled water will occur. To answer this question, our results suggest that the effects of dynamic factors on cloud phase changes should be considered in the parameterization of cloud phases within GCMs in order to further reduce the biases of climate feedbacks and climate

sensitivity among these models.

**Acknowledgments**.

This research was jointly supported by the key Program of the National Natural
Science Foundation of China (41430425), Foundation for Innovative Research Groups of the National Science Foundation of China (Grant No. 41521004), National Science Foundation of China (Grant No. 41575015 and 41205015) and the China 111 project (No. B13045). We would like to thank the CALIPSO, CloudSat and ERA-Interim science teams for providing excellent and accessible data products that
made this study possible.







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





**Table1.** Cloud phase partitioning schemes used in different climate models (cited from the table 1 of Choi et al. (2014).

| GCM | reference | Scheme[a] | $T_{ice,}$° C | $T_{w,}$° C | $n$ |
|---|---|---|---|---|---|
| ERA40 | (Weidle and Wernli (2008)) | 1 | -23 | 0 | 2 |
| CAM3 | (Collins et al. (2004)) | 1 | -40 | -10 | 1 |
| CAM5 | (Song et al. (2012)) | 1 | -35 | -5 | 1 |
| GISS, Land | (Del Genio et al. (1996)) | 2 | -40 | -10 | 2 |
| GISS, Ocean | (Del Genio et al. (1996)) | 2 | -40 | -4 | 2 |
| LMDZ(standard version) | (Doutriaux-Boucher and Quaas (2004)) | 1 | -15 | 0 | 6 |
| LMDZ(modified version | (Doutriaux-Boucher and Quaas (2004)) | 1 | -32 | 0 | 1.7 |

[a] Scheme 1: water cloud fraction $f = \left( \dfrac{T - T_{ice}}{T_w - T} \right)^n$, here $T_{ice} \leq T \leq T_w$

[a] Scheme 2: water cloud fraction $f = \exp[-(\dfrac{T_w - T}{15})^n]$, here $T_{ice} \leq T \leq T_w$











**Figure Captions**

Fig.1. The geographic and seasonal variations of $T_w$ value over 2°×6° grid boxes based on the 2B-CLDCLASS-Lidar product.

Fig.2. The geographic and seasonal variations of $T_{ice}$ value over 2°×6° grid boxes based on the 2B-CLDCLASS-Lidar product.


Fig. 3. Parameter $n$ vs the mean supercooled water cloud fraction between -40ºC to 0ºC. The color presents the numbers of grid.

Fig.4. The geographic and seasonal variations of parameter $n$ over 2°×6° grid boxes
based on the 2B-CLDCLASS-Lidar product.

Fig.5. The geographic and seasonal variations of the grid mean value of absolute difference (annual mean) between calculated and observed SCFs for different schemes, respectively. (a) for the scheme 1 used the dynamical thresholds of $T_{ice}$, $T_w$ and $n$; (b)
for the scheme 2 used the dynamical thresholds of $T_{ice}$, $T_w$ and $n$; (c) for the CAM3; and (d) for the CAM5.

Fig.6. The geographic and seasonal variations of the grid mean value of relative difference (annual mean) between calculated and observed SCFs for different schemes,
respectively. (a) for the scheme 1 used the dynamical thresholds of $T_{ice}$, $T_w$ and $n$; (b) for the scheme 2 used the dynamical thresholds of $T_{ice}$, $T_w$ and $n$; (c) for the CAM3; and (d) for the CAM5.

Fig.7. (a) The observed vertical distribution of zonal mean SCF with temperature; and
the difference of vertical distribution between calculated and observed SCFs, (b) for the scheme 1 used the dynamical thresholds of $T_{ice}$, $T_w$ and $n$; (c) for the CAM3; and (d) for the CAM5.

Fig.8. The geographic and seasonal variations of supercooled water cloud fraction at
-20ºC over 2°×6° grid boxes.

Fig.9. The geographic and seasonal variations of relative aerosol occurrence frequency (RAOF) at -20ºC over 2°×6° grid boxes based on the CALIPSO level 2 5 km aerosol level product.

Fig.10. The seasonal and zonal variations of SCF and RAOF at -20ºC, LTSS and 500hPa vertical velocity.

Fig.11. The geographic and seasonal variations of the 50% supercooled water cloud
fraction-Top temperature ($T_{50}$) over 2°×6° grid boxes.

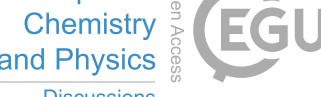



Fig.12. The dependences of $T_{50}$, $n$ and SCF at -20ºC on the RAOF and 500hPa vertical velocity. The error bars correspond to the $\pm 5$ standard error.

Fig.13. The dependences of $T_{50}$, $n$ and SCF at -20ºC on the RAOF and surface temperature. The error bars correspond to the $\pm 5$ standard error.

Fig.14. The dependences of $T_{50}$, $n$ and SCF at -20ºC on the RAOF and LTSS. The error bars correspond to the $\pm 5$ standard error.






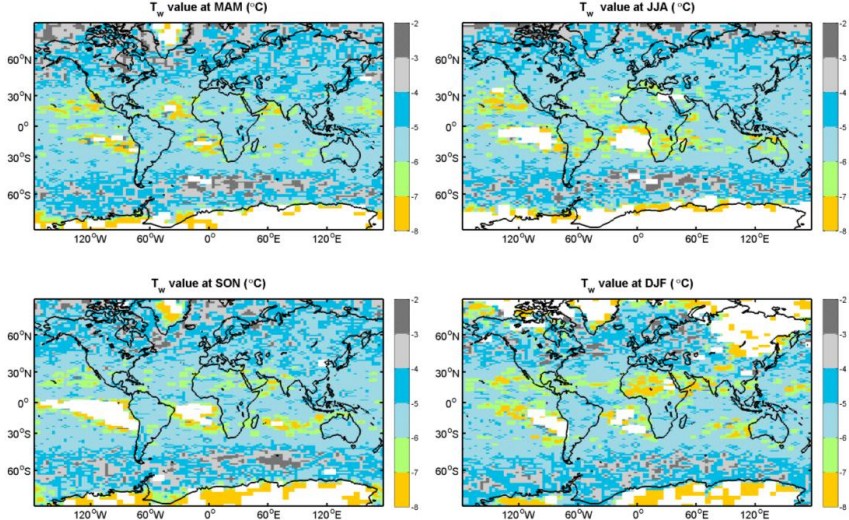

Fig.1. The geographic and seasonal variations of $T_w$ value over $2^{\circ} \times 6^{\circ}$ grid boxes based on the 2B-CLDCLASS-Lidar product.

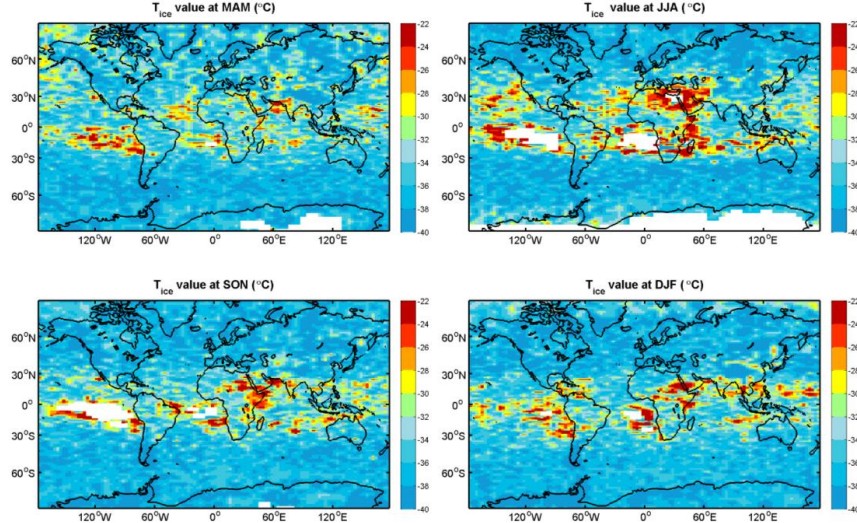

Fig.2. The geographic and seasonal variations of $T_{ice}$ value over $2^{\circ} \times 6^{\circ}$ grid boxes based on the 2B-CLDCLASS-Lidar product.





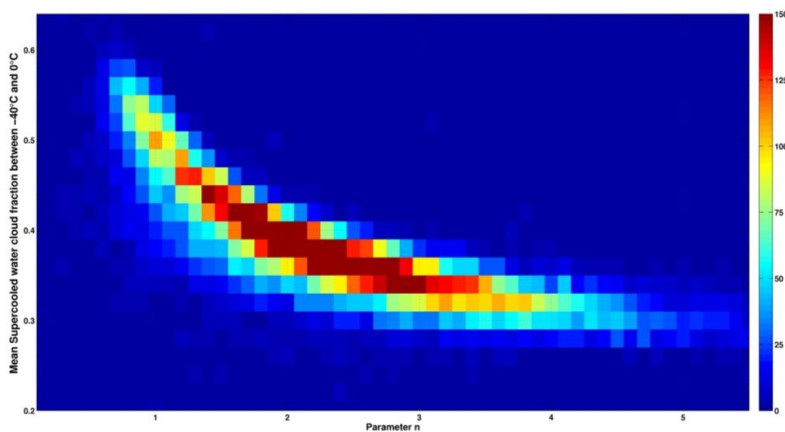

Fig. 3. Parameter *n* vs the mean supercooled water cloud fraction between -40 ℃ to 0 ℃. The color presents the numbers of grid.

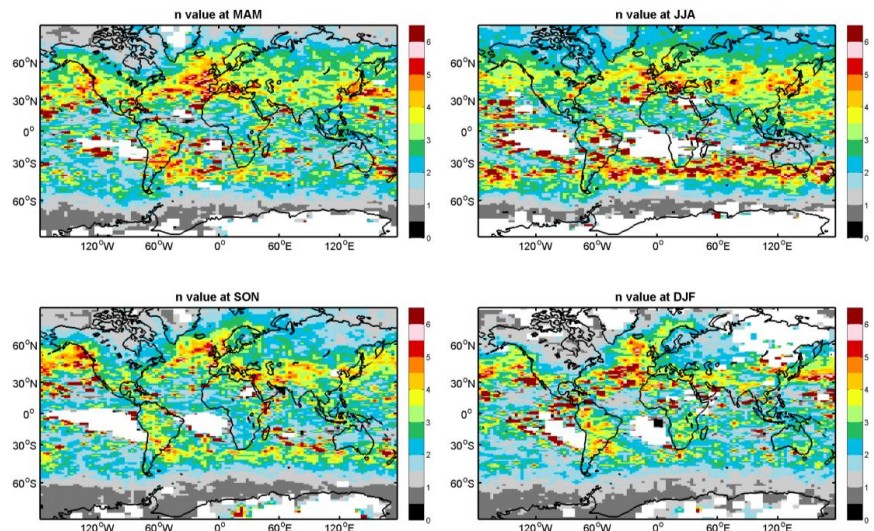

Fig.4. The geographic and seasonal variations of parameter *n* over 2°×6° grid boxes based on the 2B-CLDCLASS-Lidar product.





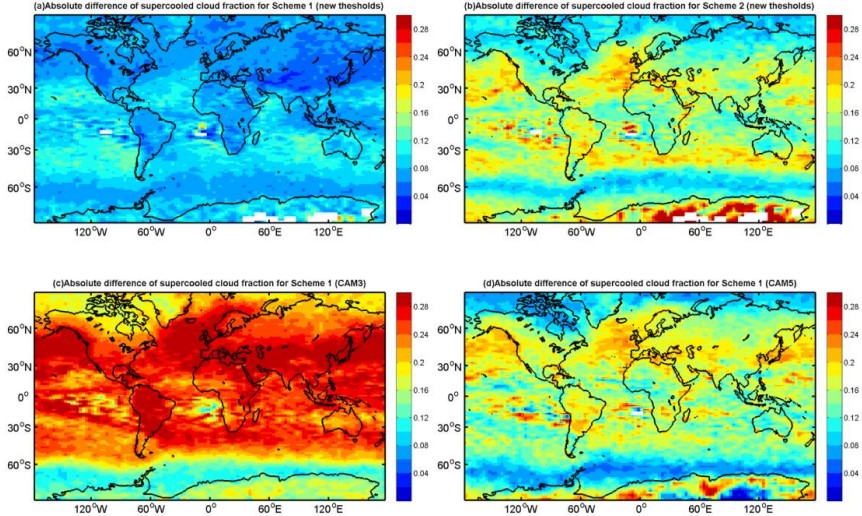

Fig.5. The geographic and seasonal variations of the grid mean value of absolute difference (annual mean) between calculated and observed SCFs for different schemes, respectively. (a) for the scheme 1 used the dynamical thresholds of $T_{ice}$, $T_w$ and $n$; (b) for the scheme 2 used the dynamical thresholds of $T_{ice}$, $T_w$ and $n$; (c) for the CAM3; and (d) for the CAM5.

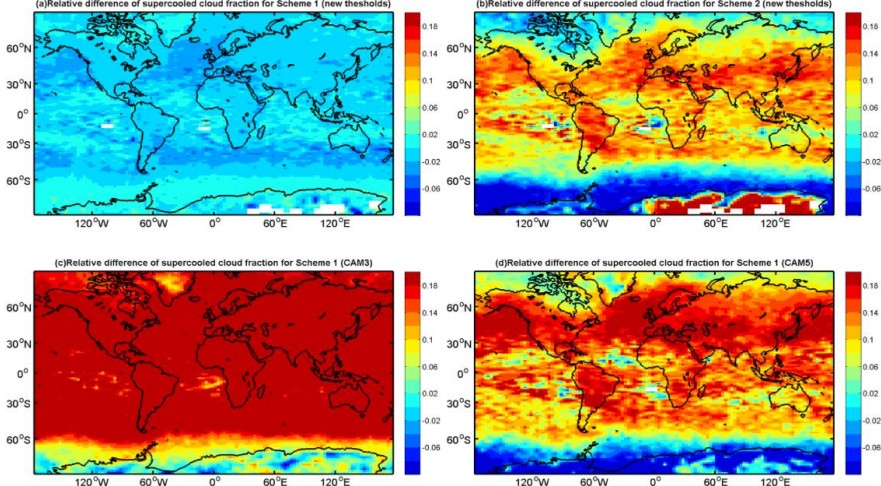

Fig.6. The geographic and seasonal variations of the grid mean value of relative difference (annual mean) between calculated and observed SCFs for different schemes, respectively. (a) for the scheme 1 used the dynamical thresholds of $T_{ice}$, $T_w$ and $n$; (b) for the scheme 2 used the dynamical thresholds of $T_{ice}$, $T_w$ and $n$; (c) for the CAM3; and (d) for the CAM5.

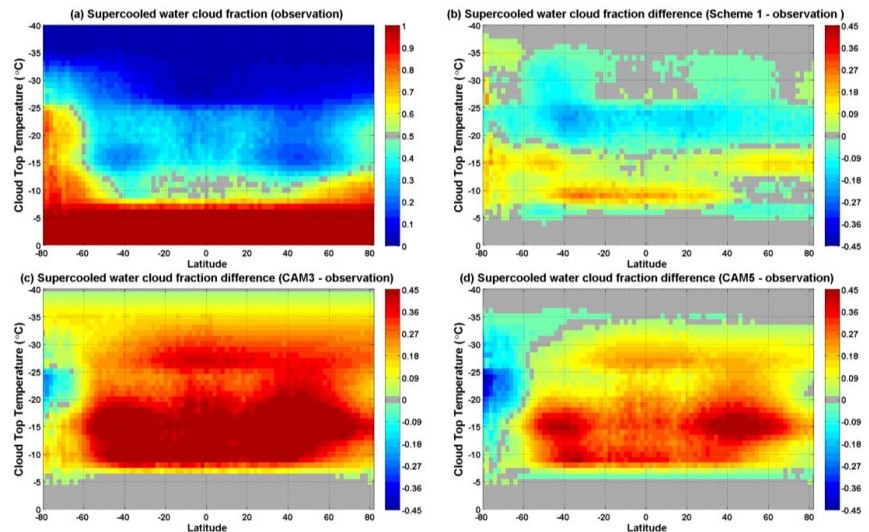

Fig.7. (a) The observed vertical distribution of zonal mean SCF with temperature; and the difference of vertical distribution between calculated and observed SCFs, (b) for the scheme 1 used the dynamical thresholds of $T_{ice}$, $T_w$ and $n$; (c) for the CAM3; and (d) for the CAM5.

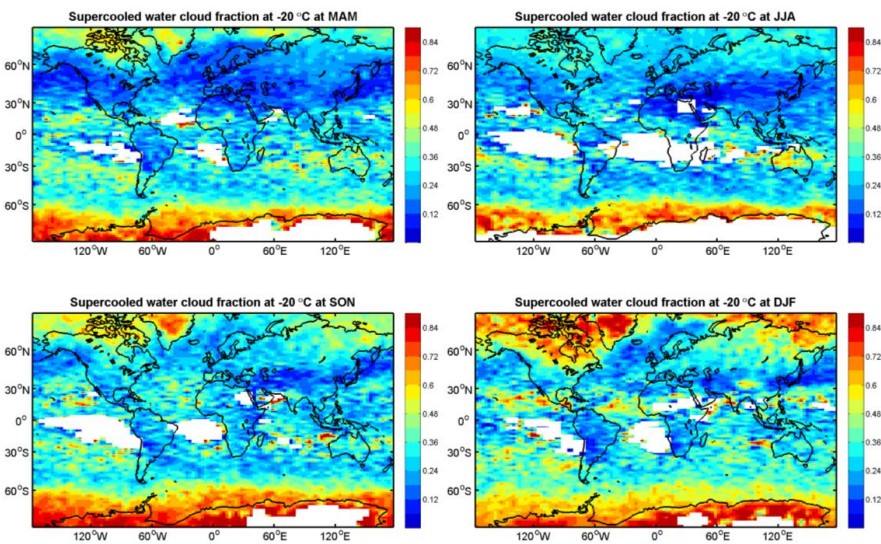

Fig.8. The geographic and seasonal variations of supercooled water cloud fraction at -20 ℃ over $2^{\circ} \times 6^{\circ}$ grid boxes.




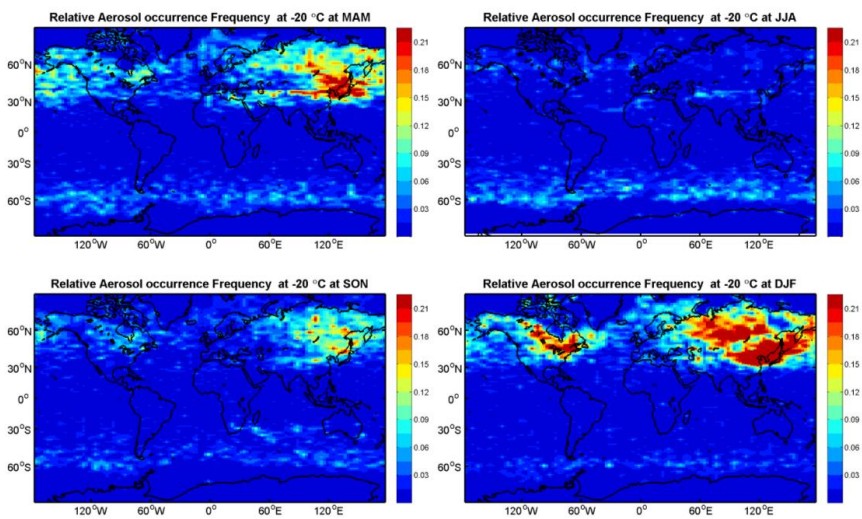

Fig.9. The geographic and seasonal variations of relative aerosol occurrence frequency (RAOF) at -20 ℃ over $2^{\circ} \times 6^{\circ}$ grid boxes based on the CALIPSO level 2 5 km aerosol level product.

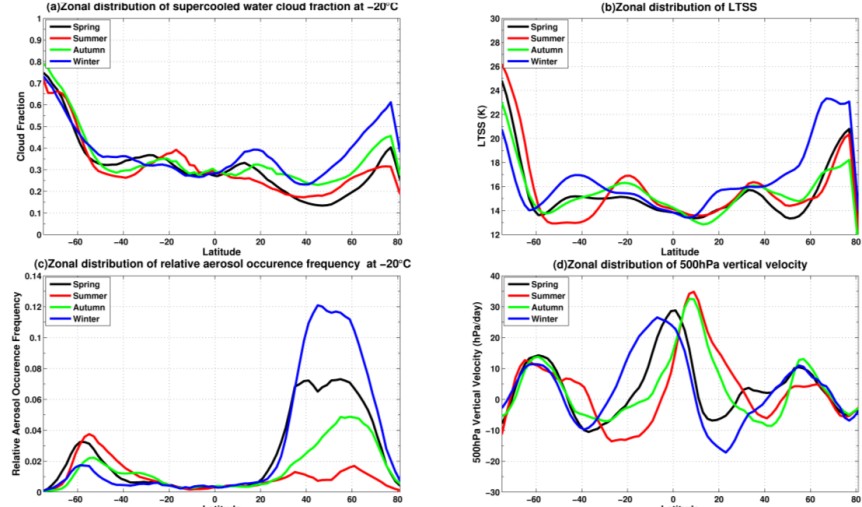

Fig.10. The seasonal and zonal variations of SCF and RAOF at -20 ℃, LTSS and 500hPa vertical velocity.



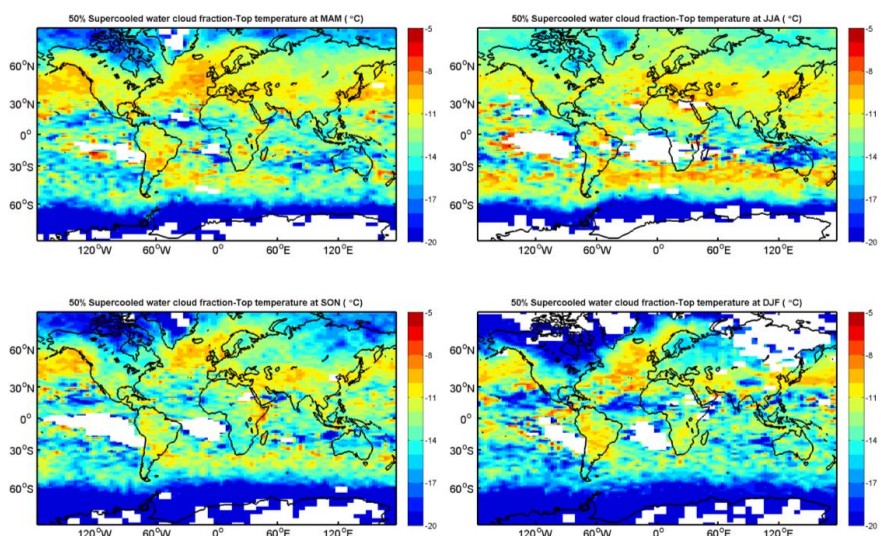

Fig.11. The geographic and seasonal variations of the 50% supercooled water cloud fraction-Top temperature ($T_{50}$) over $2^{\circ} \times 6^{\circ}$ grid boxes .

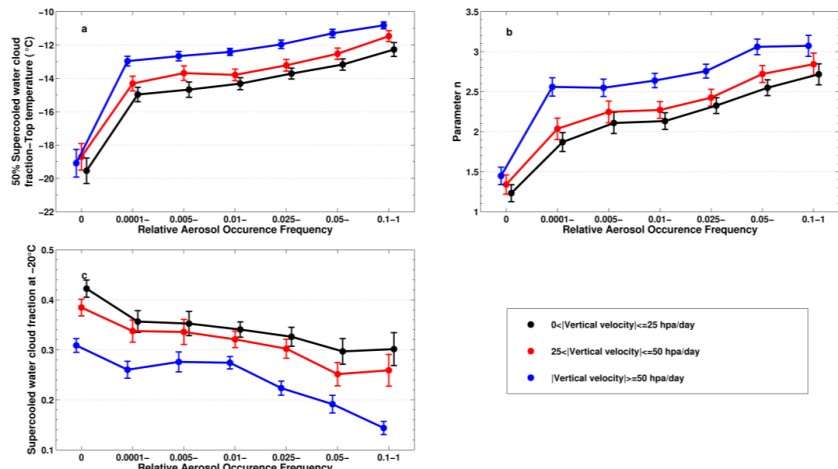

Fig.12. The dependences of $T_{50}$, $n$ and SCF at -20 ℃ on the RAOF and 500hPa vertical velocity. The error bars correspond to the $\pm 5$ standard error.





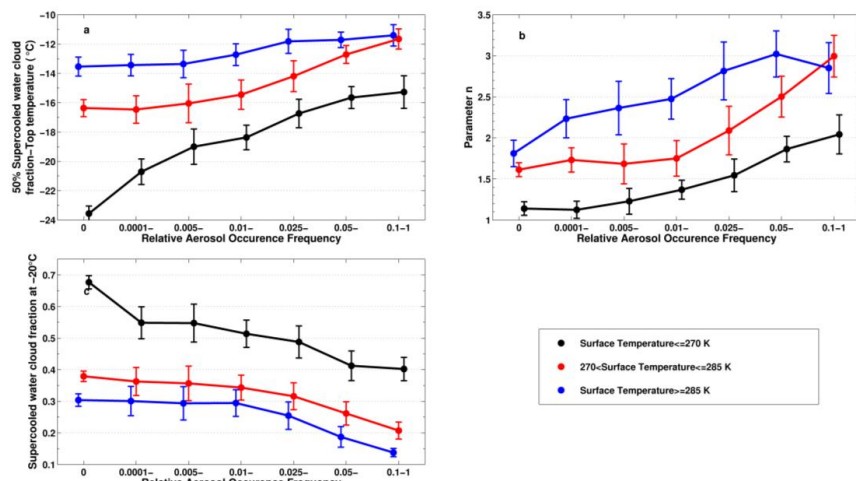

Fig.13. The dependences of $T_{50}$, $n$ and SCF at -20℃ on the RAOF and surface temperature. The error bars correspond to the ±5 standard error.

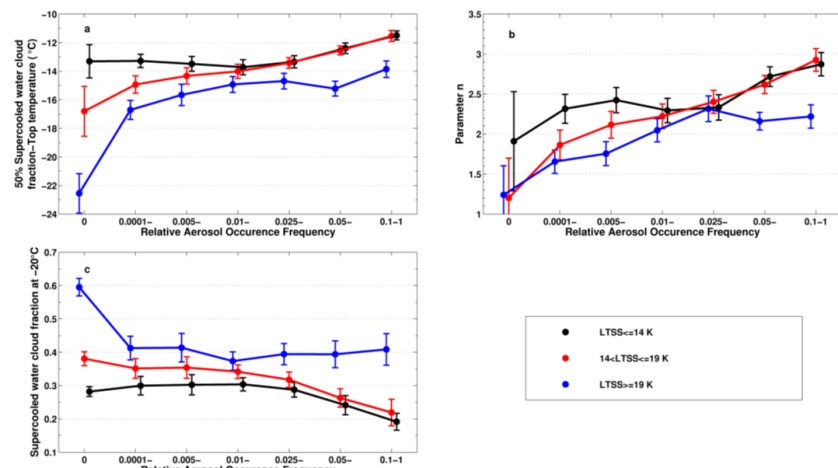

Fig.14. The dependences of $T_{50}$, $n$ and SCF at -20℃ on the RAOF and LTSS. The error bars correspond to the ±5 standard error.