# Peer review of "Effects of atmospheric dynamics and aerosols on the fraction of supercooled water clouds"

_Atmospheric Chemistry and Physics, 2016_

## Referee Comment (RC1) · Anonymous Referee #1 · 22 Mar 2016

In this paper, the authors mostly analyze the geographical and seasonal variation of the relation between temperature and supercooled-liquid cloud frequency based on CloudSat 2B-CLDCLASS-LIDAR product. To this end, they utilize different type of equation and sets of parameters and find which one better represents the observations. Then, they investigate the effect of aerosols, vertical velocity at 500 hPa, Lower Tropospheric stability and sea surface temperature on the transition temperature between liquid-dominated and ice-dominated clouds in mixed-phase clouds (0degC<T<-40degC). Although the last part of the study may contain some interesting results if further developed and better presented, the first part of the study - which represents half of the results - is not representative of the title of the study and does provide any new insights on the topic of supercooled-liquid clouds and/or mixed-phase clouds that has already been published (Cesana and Chepfer, 2013 JGR; Cesana et al., 2015

JGR; McCoy et al., 2015 JGR; Tan and Storelvmo, 2015 JAS; Yoshida et al., 2010 JGR). Indeed, the authors missed a lot of recent publications on that field, which has received more and more attention lately. However, the authors, by using an independent method based on CloudSat, confirm previous results obtained by using CALIPSO observations (mentioned previously). But in its actual format, it is not sufficient to be published in a scientific journal and may be more relevant for technical study/report or validation method paper by comparison to CALIPSO products. This is why I strongly encourage the authors to remove the comparison with "model relation" part and to focus their study on the observational part in case they want to resubmit their work.

Overall, a lot of statements are vague and/or confused and are not supported by any references. Later on, numbers come from nowhere and seem to be guessed more than calculated. I strongly encourage the authors to clearly state whether they computed numbers or just guessed. Several results and conclusions only rely on visual inspections and hypothesis (qualitative analysis) rather than actual quantitative evidences (such as correlation or regression etc), particularly in the last part. In conclusion,. In addition, the authors should show more numerical results in terms of means/correlations to strengthen our confidence in the results rather than just showing map and doing qualitative analysis based on those maps.

Other major flaws stand out in the paper. For example, the authors can't compare modeled Temperature-Phase relationship directly with that observed for several reasons (e.g. Cesana and Chepfer 2013, Fig. 11; Cesana et al., 2015, Fig. 1). It does not take into account: i) The bias of the instrument: the lidar cannot pass through optically thick clouds making the relation valid only for certain clouds. The lidar is more sensitive to liquid droplets than ice crystals in mixed phase clouds, which affects the shape of the T-Phase relation. ii) The sampling (spatio-temporal) effects iii) The different cloud/cloud phase definitions: The observations of SCF are liquid/ice frequency of occurrence ratio whereas the modeled SCF are based on ice/liquid water content mass ratio. Besides, you only select the upper part of the cloud whereas the relation

is used in the whole column in the models. For all the above reasons, the modeled T-Phase relation cannot be compared directly to observations like it is done in the paper. If the authors really want to compare with observations, they'll have to either use a simulator of the instruments on the model (e.g. Cesana and Chepfer 2013) or ensure the comparison is possible and consistent by choosing conditions that reduce the differences cited above (e.g. Cesana et al., 2015). The figure 7a is quite different from what published in Yoshida et al (2010, Fig. 6), Hu et al. (2010, Fig. 7) and Cesana et al (2015, Fig. 5) showing the importance of using comparable datasets to evaluate the models. The also show that the T-Phase relation varies depending on the latitude and thus regionally by extension. Besides, Cesana and Chepfer (2013, fig. 11) showed specifically the regional variation of the T-Phase relation – while existing – was small.

Moreover, the authors insist in the fact that most models only use temperature-dependent relation to determine their cloud phase. This is clearly not the case anymore. Cesana et al. (2015) have shown that 5 out of 16 models of their study used the temperature as unique criterion to determine the cloud phase. And the ones using the temperature only are currently working on new schemes. Finally, CAM5 model is only using a T-Phase relation for the convective detrainment and not everywhere as stated in the paper. Besides, the relation mentioned in the paper for CAM5 is not correct, the parameters are Tw=-10degC and Tice=-40degC.

Regarding the previous general comments, I don't recommend this paper for publication in ACP. However, I strongly encourage the authors to work on the later part (relation of the cloud phase transition with the aerosols) of the paper and to resubmit another more focused manuscript including more quantitative results.

Specific comments:

Line 54: -30degC

Line 86: "However... climate models." This sentence is too vague and the 2 parts are not really connected. The authors should reformulate and specify what kind of

observations (satellite insitu? All?), what kind of processes? (macro,micro?). Besides, Klein et al., 2013 and Zhang et al., 2005 do not refer to climate change/future climate but to present/past climate simulations. The authors should remove these 2 references.

Lin 90:"One of... ni GCMs." I don't know where the authors can find a list of the primary challenges but these study are quite old and do not represent the current primary challenges. I strongly recommend changing this sentence. Yet, I believe the supercooled liquid clouds and mixed-phase clouds are crucial to reduce the climate feedbacks uncertainties, as shown in McCoy et al., 2015.

Line 95: The authors use the term currently and refer to 2 studies that use old models. Cesana et al., 2015 and McCoy et al., 2015 (the list is not exhaustive) are more recent papers that illustrate this statement.

Line 104: Can the authors reference studies here? (e.g. Forbes et al., 2014 MWR)

Line 107: References are missing for CC theory and the laboratory results.

Line 110: The authors should mention in situ studies that are the most "trustable" observations (e.g. Heymsfield and Miloshevich, 1993, JAS)

Line 126: This "exponent" has not been defined. Please define it or remove the sentence.

Line 130: This sentence is difficult to understand and most likely not grammatically correct. Please reformulate.

Line 153: They only defined a relation between the cloud top temperature and the supercooled liquid fraction based on a best fit of the observations, which is very different from a model "parameterization". Please, change the last part of the sentence as well as the next 2 sentences.

Line 215: I'm assuming the authors are talking about the high-latitude mixed-phase clouds with a supercooled-liquid layer on top and precipitating ice below. However, the

other way around may also happen, with an ice-topped layer. So please clarify.

Line 285: The model used in Doutriaux-Boucher and Quaas (2004) is obsolete and Hu et al. (2010) is not a model-based study. Please, use more recent references.

Eq (1) is wrong; T at the denominator should be Tice (also in Table 1)

Line 296: It is not between -40 and 0degC but between Tice and Tw

Although CAM5 partially uses temperature "ramp" (in convective detrainment), it uses most of the time prognostic equations to calculate liquid and ice mixing ratios. This T-phase equation is therefore not representative of the cloud phase in CAM5. Moreover, the standard version uses Tice = -40degC and Tw = -10degC rather than -35degC and -5degC used in the modified version of Song et al (2012, journal of climate). The authors should mention this somewhere in the manuscript.

In addition, the new ERA and LMDZ models use slightly different T-phase relations now. Finally, is there a reason to choose these specific relations out of the Choi paper?

Line 297-302: This sentence is too long and the statement is not really supported by any references.

Line 302-305: Same comment, no references to support these facts that could be the topic of a whole paper (e.g. Tan and Storelvmo, 2015; McCoy et al., 2015).

Line 320: Actually, strong subsidence may contribute to dissipate stratocumuli. The weak subsidence favors stratocumulus formation (Wood et al., 2012, MWR).

Eq 3 and 4 are the same. I guess you forgot to remove the /41 in eq 3.

Line 409: Reference?

Line 427: I strongly encourage the authors to be more rigorous when they mention numbers. For example, Tice does not seem to be -35degC judging from the figure 7a.

Line 441: It is not CAM3 or CAM5 but the T-phase relation that shows over or underestimation of SLF.

Line 457-459: The analysis does not demonstrate this at all. It just shows that the T-Phase relation based on the 2B-CLDCLASS-LIDAR product is different from those used in some models, which was very much expected. However, the inability of GCMs to reproduce observed features of the cloud phase is not new and has been already demonstrated in previous studies using actual GCMs output rather than just the temperature-cloud phase relation (Chen et al., 2012; Cesana and Chepfer, 2013; Komurcu et al., 2014 JGR; Cesana et al., 2015; Tan and Storelvmo, 2015; the list is not exhaustive).

Line 460-462: Reference?

Why did the authors choose -20degC. If there is a special reason, please explain, otherwise it would be worth to check the sensitivity of other temperature isotherms.

Line 467: Is it a guess based on visual inspection or did the authors actually calculate the numbers?

This part is unclear and confusing. I don't see how the Fig. 8 verifies the later statement that changes in the SCF are correlated to dust. The following sentence is also unclear.

Line 477: The authors can't conclude this just based on 2 maps at -20degC without even looking for a statistical correlation between SLF and aerosols. A better way would be to focus on a specific region and study the SLF depending on the aerosol load.

The last part is very confusing and could be squeezed easily. Also I don't understand the absolute value for the vertical velocity, which is very confusing because we expect different results from positive or negative vertical velocity. Besides, the authors should define what positive vertical velocity means somewhere because in GCM studies, positive generally mean subsidence.

Finally, in fig. 12, the difference between T50 at 0 and 0.0001 (% ???) of aerosol frequency seems to be an artifact rather than a real observation and does not mean
than aerosol have more influence than vertical velocity.

---

## Referee Comment (RC2) · Anonymous Referee #2 · 23 Mar 2016

article

**1   General comments**

This study presents statistical relationships between cloud-top cloud thermodynamic phase and aerosols (dust, polluted dust and smoke) as well as meteorological variables (vertical velocity at 500 hPa, lower tropospheric static stability (LTSS) and surface temperature) to infer the influence of atmospheric dynamics on cloud thermodynamic phase using a combination of global satellite observations and reanalysis data over a 4-year period (2007-2010). The authors first evaluate the cloud thermodynamic phase partitioning schemes in a handful of models against observations. They find that out of

the models, the cloud thermodynamic phase partitioning schemes in CAM3 and CAM5 compare best to the observations used in their study. The authors then proceed to show that vertical motions can explain the seasonal cycle of supercooled water cloud fraction (SCF) in regions where aerosols cannot explain the seasonal cycle of SCF. They find that strong vertical motions appear to be correlated with regions of low SCF likely through an enhanced precipitation rate, and that higher LTSS appears to be correlated with regions of high SCF. This work presents interesting results that could be useful for near-future model development, however, substantial revisions pertaining to the content, quality and writing style of the manuscript should be undertaken. Specific comments are provided below.

**2  Specific Comments**

1. **Title:** The study could be separated into two parts, the first part evaluating the temperature ramp schemes used in climate models against observations and the second part examining statistical relationships between dynamical variables and SCF. The title only reflects the latter part. Please change the title to better reflect the content of the manuscript.

2. **Introduction:** The logical flow can be improved to enhance clarity. Cold cloud schemes in models are discussed in the first paragraph before the existence of supercooled liquid clouds in the second paragraph. Also, on lines 107-109: the Clausius-Clapeyron equation simply relates the saturation vapour pressure and the temperature. If the authors wish to cite theoretical support for the existence of liquid, they should refer to the free energy barrier of pure water droplets and classical nucleation theory.

3. **Datasets and Methods:**

[Figure]

- Lines 177-185: Please include indicate that the ERA-Interim reanalysis dataset was used to obtain the aerosol and cloud-top temperatures.

- Line 182: why was a resolution of $2° \times 6°$ chosen? The longitude dimension is quite wide. Please clarify.

- Line 184: It's not clear to me why only daytime observations were used. Wouldn?t it be better to use nighttime observations, especially for the CALIOP observations since sunlight decreases with the signal to noise ratio?

4. **Results**: Section 3.1 is not a result. This section is more appropriate for Section 2 (Datasets and Methods). Also, much of the beginning paragraph in this section that describes the lack of dependency of cloud thermodynamic phase on ice nucleation in model schemes is redundant with what was already written in the introduction and does not need to be repeated.

- Fig. 1: It would be more helpful to distinguish no data regions from regions where SCF is not unity at temperatures below $0°C$.

- Fig. 5: The authors use panels a and b in this figure to demonstrate that scheme 1 is more accurate than scheme 2 in terms of simulating SCF compared to the observed values, but could it not be interpreted from panels c and d, which use scheme 1, that scheme 1 can do even more poorly than scheme 2 depending on what the dynamical thresholds of $T_{ice}$, $T_w$ and n are? The authors have mentioned that $T_{ice}$ is "unreasonably" low $(-40°C)$ in CAM3, which I assume implies that this could explain why CAM3 does poorly even with scheme 1, but scheme 2 also has the same $T_{ice}$ $(-40°C)$ and predicts smaller absolute differences than CAM3. Please clarify.

- Equations 3 and 4 (and lines 384-385): If each bin is $1°C$ and there are bins from $0°C$ to $-40°C$ then shouldn?t there be 40 bins (not 41)?

- Figures 3 and 4 do not contain any information about the vertical distribution of n. Please consider including this information in an additional plot, as it may be useful to provide this information to the readers.

- Why have the global distributions of the vertical velocity at 700 hPa, LTSS and surface temperature have not been plotted? It may help to plot these since Figures 12, 13 and 14 do not contain any information about the distribution of these variables. Also, have pattern correlation coefficients between the variables been calculated?

- Fig. 6: Please consider using a more intuitive colour bar, i.e. positive values in a red gradient, negative values in a blue gradient, and zero values in white (or grey as in Fig. 7).

- Figure 14b: Why does greater the case when LTSS is less than or equal to 14 K not result in a higher n value for the bins with higher relative aerosol occurrence frequency?

5. Probably my biggest concern about the manuscript is that the model cloud thermodynamic phase partitioning schemes in Table 1 may not be directly comparable to the cloud-top observations made by CloudSat and CALIPSO in this study. The CAM3 and CAM5 schemes, at least are not, since the temperatures do not refer to the cloud-top temperatures and these limitations should be discussed in the text. Having said that, the conclusions that the authors have drawn regarding the realism of the cloud thermodynamic phase partitioning schemes would only be true if these schemes are fully consistent with how a satellite would observe the clouds, i.e. from the perspective of a satellite simulator. The authors have not run any model simulations in this study, and may find that even though the general formula of the schemes in Table 1 agree well with observations, that the actual model-computed SCFs may not agree very well with the observations after all since they are not comparing apples to apples in a strict sense. Furthermore, the authors should note that the temperature ramp used in CAM5 given in Table

1 is specifically for detrained convective condensate. Liquid and ice mass and number concentrations for stratiform clouds are computed from prognostic equations in CAM5, which has a very different cloud microphysics scheme from that in previous version (e.g. CAM3/CAM4). This may also be the case for the other models. Please discuss these points.

6. The goal of many climate models is to move away from temperature ramp schemes in general, such as those for cloud thermodynamic phase partitioning listed in Table 1. This is unlikely to be accomplished any time soon, though, and the work of the authors in this respect is useful for the modelling community. However, the authors should discuss the move toward prognostic schemes in climate models, which many have already adopted.

7. Lines 740-741: This sentence is a bit ambiguous. There is evidence suggesting that a cloud phase feedback occurs, causing more shortwave to be reflected back out to space relative to the state prior to global warming. This finding can be briefly discussed here. A few references relating to the cloud phase feedback and cloud thermodynamic phase repartitioning are listed below:

   - Mitchell, J.F.B., Senior, C. A., and Ingram, W. J. $CO_2$ and climate: a missing feedback? Nature, 341, 132-134, 1989.
   - McCoy, D. T., Hartmann, D. L., Grosvenor, D. P. Observed Southern Ocean Cloud Properties and Shortwave Reflection. Part II: Phase Changes and Low Cloud Feedback. Journal of Climate, 27, 8858-8868, 2014.
   - McCoy, D. T., Hartmann, D. T., Zelinka, M. D., Ceppi, P., Grosvenor, D. P., Mixed?phase cloud physics and Southern Ocean cloud feedback in climate models. Journal of Geophysical Research: Atmospheres, 120, 9539-9554, 2015.
   - Storelvmo, T., Tan, I., Korolev, A. V. Cloud phase changes induced by $CO_2$ warming — a powerful yet poorly constrained cloud-climate feedback. Curr.

Clim. Change Rep., 1(4), 288–296, 2015.

- Tsushima et al. (already in the references).

- Finally, I very strongly recommend that the authors ask a native English speaker to proofread for grammatical errors.

**3 Technical Corrections**

- Abstract: Please indicate that the aerosols refer specifically to dust, polluted dust and smoke aerosols here.

- Please enlarge the fonts in all figures and include only high-resolution plots created using vector graphics.

- Lines 177: The word "current" is preferred over "following".

- Line 190: CPR was already defined on line 144.

- Fig. 3 caption: Technically, what's plotted is the mean supercooled water cloud fraction vs. the parameter n (what's on the abscissa), not the other way around. However, n is the dependent variable here since it is fitted based on what f is, so it would be better to have it on the ordinate.

- Line 474: Please clearly define the relative aerosol occurrence frequency.

- Line 486: The sentence is missing a punctuation mark at the end of it.

- Fig. 10: Please change the title of the ordinate to "Supercooled Water Cloud Fraction" in panel a (not to be confused with the total cloud fraction or any other type of cloud fraction). Also, please specify that these seasons refer to the northern hemisphere.

- Line 499: I think what is meant is "consistency", not "inconsistency".

- Line 612: Insert "that" between "everywhere" and "the".

---

## Author Comment (AC1) · 3 Jun 2016

**Response to Reviewer #1's Comments:**

**Jiming Li et al. (Author)**

**We are very grateful for the Review #1's for pointing out a number of weaknesses and addressing significant comments on the original manuscript, which are very helpful and have led to significant improvements of this paper. Based on Reviewer #1's comments, we rewrote the manuscript and paid more attentions to investigate the impacts of meteorological parameters on the supercooled liquid cloud fraction under different aerosol loadings at a global scale. In addition, some superfluous information in each section was deleted and some interpretations in each section were added in order to make the manuscript more clear. Some grammatical errors already were corrected in the revision and the paper also be edited by a native English speaker to make it more readable.**

**Detailed information:**

(1)Due to the modeled T-Phase relation cannot be compared directly to observations like it is done in the paper, Reviewer #1 suggested us to remove the comparison with "model relation" part and focused our study on the observational part (relation of the cloud phase transition with the aerosols). We very thank reviewer for pointing out the major flaws of this paper and providing some important explanations about these flaws. In the revised paper, we followed the suggestion from reviewer #1 to remove the comparison part with "model relation". In addition, duo to some studies have investigated the impact of different aerosol types on cold phase clouds over East Asia (Zhang et al., 2015) or at a global scale (Choi et al., 2010; Tan et al., 2014). However, systematic studies of the statistical relationship between cloud phase changes and meteorological parameters at a global scale have received far less attention. Thus, the revised paper paid more attentions to investigate the impacts of meteorological parameters on the supercooled liquid cloud fraction at a global scale.

(2) We reorganized the introduction section. Some confused sentences and wrong quotations were revised.

(3) In the section 2, we replaced the cloud phase information from the 2B-CLDCLASS-LIDAR product with the GCM-Oriented Cloud-Aerosol Lidar and Infrared Pathfinder Satellite Observation (CALIPSO) Cloud Product (GOCCP). This product can provide us more longer-time cloud phase information. Thus, all statistical relationship in the revised paper were derived from 8 years (2008–2015) of data from CALIPSO-GOCCP, the ERA-Interim daily product and the CALIPSO level 2, 5 km aerosol layer product. Some introductions about datasets were added in this section. Please see the section 2.

(4) In the section 3 (results part), we did a lot of changes, and mainly investigated the temporal correlations over the 8-year period (96 months) between monthly supercooled water cloud fraction and different meteorological parameters. Some new results were added. For those regions with temporal correlations between SCFs and meteorological parameters at the 95% confidence level were further used to calculate the spatial correlations between SCFs and meteorological parameters.

**Specific responses**

We appreciated the insightful suggestion and comments made by reviewer. In the revised paper, the comparison with "model relation" part was removed. Thus, we only provided the point-by-point responses to the reviewer's comments about the observational part.

**(1)** Line 104: Can the authors reference studies here? (e.g. Forbes et al., 2014 MWR)

**Response:** In revised paper, we added this reference in the introduction section. In addition, some related latest studies also were added.

**(5)** Why did the authors choose -20degC. If there is a special reason, please explain, otherwise it would be worth to check the sensitivity of other temperature isotherms.

**Response:** We agreed with reviewer. In the revised paper, some statistical results at other isotherms (such as -10°C and -30°C) also are analyzed and summarized (see the Table 1).

**(6)** Line 477: The authors can't conclude this just based on 2 maps at -20degC without even looking for a statistical correlation between SLF and aerosols. A better way would be to focus on a specific region and study the SLF depending on the aerosol load. The last part is very confusing and could be squeezed easily. Also I don't understand the absolute value for the vertical velocity, which is very confusing because we expect different results from positive or negative vertical velocity. Besides, the authors should define what positive vertical velocity means somewhere because in GCM studies, positive generally mean subsidence.

**Response:** We appreciated the insightful suggestions and comments. In the revised paper, we added the statistical correlation between SLF and different meteorological parameters by performing the temporal and spatial correlation analysis. We found that same meteorological parameter has a distinct effect in different regions on the SCFs. Please the section 3.2 and 3.3 of revised paper.

---

## Author Comment (AC2) · 3 Jun 2016

**Response to Reviewer #2's Comments:**

**Jiming Li et al. (Author)**

**We are very grateful for the Review #2's detailed comments and suggestions, which help us improve this paper significantly. Some grammatical errors already were corrected in the revision and the paper also be edited by a native English speaker to make it more readable. Based on two Reviewers' comments, we rewrote the manuscript and paid more attentions to investigate the impacts of meteorological parameters on the supercooled liquid cloud fraction under different aerosol loadings at a global scale. In addition, some superfluous information in each section was deleted and some interpretations in each section were added in order to make the manuscript more clear.**

**Detailed information:**

(1)Due to the modeled T-Phase relation cannot be compared directly to observations like it is done in the paper, Reviewer #1 suggested us to remove the comparison with "model relation" part and focused our study on the observational part (relation of the cloud phase transition with the aerosols). In the revised paper, we followed the suggestion from reviewer #1 to remove the comparison part with "model relation". In addition, duo to some studies have investigated the impact of different aerosol types on cold phase clouds over East Asia (Zhang et al., 2015) or at a global scale (Choi et al., 2010; Tan et al., 2014). However, systematic studies of the statistical relationship between cloud phase changes and meteorological parameters at a global scale have received far less attention. Thus, the revised paper paid more attentions to investigate the impacts of meteorological parameters on the supercooled liquid cloud fraction at a global scale.

(2) We reorganized the introduction section. Some confused sentences and wrong quotations were revised.

(3) In the section 2, we replaced the cloud phase information from the 2B-CLDCLASS-LIDAR product with the GCM-Oriented Cloud-Aerosol Lidar and

Infrared Pathfinder Satellite Observation (CALIPSO) Cloud Product (GOCCP). This product can provide us more longer-time cloud phase information. Thus, all statistical relationship in the revised paper were derived from 8 years (2008–2015) of data from CALIPSO-GOCCP, the ERA-Interim daily product and the CALIPSO level 2, 5 km aerosol layer product. Some introductions about datasets were added in this section. Please see the section 2.

(4) In the section 3 (results part), we did a lot of changes, and mainly investigated the temporal correlations over the 8-year period (96 months) between monthly supercooled water cloud fraction and different meteorological parameters. Some new results were added. For those regions with temporal correlations between SCFs and meteorological parameters at the 95% confidence level were further used to calculate the spatial correlations between SCFs and meteorological parameters.

**Specific responses**
We appreciated the insightful suggestion and comments made by reviewer #2. In the revised paper, the comparison with "model relation" part was removed. Thus, we only provided the point-by-point responses to the reviewer's comments about the observational part.

**1. Title:** The study could be separated into two parts, the first part evaluating the temperature ramp schemes used in climate models against observations and the second part examining statistical relationships between dynamical variables and SCF. The title only reflects the latter part. Please change the title to better reflect the content of the manuscript.
**Response:** We agreed with reviewer. In the revised paper, we focused on the statistical relationship between cloud phase changes and meteorological parameters at a global scale. Thus, the title can reflect the content of the revised manuscript.

**2. Introduction:** The logical flow can be improved to enhance clarity. Cold cloud schemes in models are discussed in the first paragraph before the existence of

supercooled liquid clouds in the second paragraph. Also, on lines 107-109: the Clausius-Clapeyron equation simply relates the saturation vapour pressure and the temperature. If the authors wish to cite theoretical support for the existence of liquid, they should refer to the free energy barrier of pure water droplets and classical nucleation theory.

**Response:** We appreciated the insightful suggestion from reviewer #2. In the revised paper, we reorganized the introduction section. Some confused sentences and wrong quotations were revised.

**3. Datasets and Methods:** Lines 177-185: Please include indicate that the ERA-Interim reanalysis dataset was used to obtain the aerosol and cloud-top temperatures.

**Response:** Some detailed introductions about datasets were added in this section. Please see the section 2.

**Line 182:** why was a resolution of 2_ _ 6_ chosen? The longitude dimensionis quite wide. Please clarify.

**Response:** In the revised paper, we performed the temporal correlation between supercooled water cloud fraction and meteorological parameters. However, due to the 16-day orbit of CALIOP, the horizontal resolution of the data set had been reduced to 10 ◦ latitude by 10° longitude grid boxes to avoid the issue of a sparse data set when performing the temporal correlations, similar with the study of Tan et al. (2014).

**Line 184:** It's not clear to me why only daytime observations were used. Wouldn't it be better to use nighttime observations, especially for the CALIOP observations since sunlight decreases with the signal to noise ratio?

**Response:** Yes, we very agreed with reviewer. To avoid artifacts due to noise from scattering of sunlight, it is better to conduct the CALIOP retrieval during nighttime. However, in view of the lack of CALIPSO observations at high latitudes of the northern Hemisphere during boreal summer nights, this study utilizes the mean values of SCFs, meteorological parameters and RAFs during daytime and nighttime to perform the temporal and spatial correlations analysis.

**4. Results:** Why have the global distributions of the vertical velocity at 700 hPa, LTSS and surface temperature have not been plotted? It may help to plot these since Figures 12, 13 and 14 do not contain any information about the distribution of these variables. Also, have pattern correlation coefficients between the variables been calculated?

**Response:** We very agreed with reviewer. The temporal and spatial correlations between supercooled water cloud fraction and meteorological parameters was performed in the revised paper. In addition, we also provided the global distributions of vertical velocity at 500 hPa, LTSS, skin temperature, and u wind at 100hPa in the Fig. s1 in the supplemental materials.

**5.** "Probably my biggest concern about the manuscript is that the model cloud thermodynamic phase partitioning schemes in Table 1 may not be directly comparable to the cloud-top observations made by CloudSat and CALIPSO in this study. The CAM3 and CAM5 schemes, at least are not, since the temperatures do not refer to the cloud-top temperatures and these limitations should be discussed in the text....................Liquid and ice mass and number concentrations for stratiform clouds are computed from prognostic equations in CAM5, which has a very different cloud microphysics scheme from that in previous version (e.g. CAM3/CAM4). This may also be the case for the other models. Please discuss these points".

**Response:** Yes, **t**he modeled T-Phase relation cannot be compared directly to observations like it is done in the paper. We appreciated the insightful suggestion from reviewer #2. Based on the suggestion from the Reviewer# 1, we removed the comparison part with "model relation" and paid more attentions to investigate the impacts of meteorological parameters on the supercooled liquid cloud fraction under different aerosol loadings at a global scale.

**Line 474:** Please clearly define the relative aerosol occurrence frequency.
**Response:** The detailed information was added in section 2.3.

---

## Referee Report (RR1)

**1  General comments**

The authors have made substantial changes to the original manuscript that have eliminated much of the questionable material pertaining to the evaluation of cloud thermodynamic phase temperature ramp schemes in models. They have also expanded the latter part of the manuscript pertaining to the statistical relationships between meteorological parameters and cloud thermodynamic phase, focusing on particular regions of interest. The results are intriguing, however, my main concern is that the physical interpretations of the many correlations in the manuscript are lacking. Moreover, many parts of the analysis appear to be speculative rather than rigorously verified. I also question the robustness of some of the results, which need additional clarification. Finally, many statements in the manuscript read as though correlation implies causation and care must be taken to avoid such statements, as there could clearly be other confounding factors involved. My recommendation is: publish with major revisions. Specific comments follow.

**2  Major Comments**

1. **Results, Section 3.1, Lines 354-356:** The authors write that the negative correlations between skin temperature and SCF in the mid- and high latitudes "mean" that high skin temperature promotes the glaciation of supercooled droplets, and that it the positive correlations "mean" that high skin temperature inhibits glaciation in the tropics. Firstly, correlation does not imply causation. These correlations may be caused by many different confounding factors that lead to mechanisms that are completely unrelated to skin temperature. What is the physical mechanism for why high skin temperature promotes glaciation in the mid- and high latitudes? This was never mentioned in the manuscript. The authors explain that high skin temperature in the tropics inhibits glaciation because it triggers deep convection, which lofts the liquid droplets to the colder isotherms, but what is the mechanism for the mid- and high latitudes? Secondly, the area of the tropics that the authors are referring to appears to be only a small part of the entire tropics (how many gridboxes/what is the percentage of the area of the tropics that is positively correlated?). Thirdly, if high skin temperature is indeed inhibiting glaciation due to a triggering of deep convection in the tropics, then shouldn't the coldest isotherm, i.e. $-30°C$ also show positive correlations in the tropics as well? The correlations between relative humidity and SCF and vertical velocity and SCF appear to support your argument in the tropics, but please clarify the effect of skin temperature.

   Next, there is an apparent contradiction between the authors' explanation of the correlations found in the tropics and Figures 1, 2 and 3. The authors explain that vigorous convective activity, high relative humidity as well as high vertical velocities in this region cause supercooled liquid to loft to

higher altitudes too quickly to allow for glaciation of supercooled liquid, which would suggest that SCFs should be higher in this region for the aforementioned reasons. Yet, Figs. 1, 2 and 3 all show that the tropics contain mixed-phase clouds with some of the lowest SCFs in the world. Please clarify.

2. **Abstract, Lines 57-59:** This is another example of a sentence that reads as if correlations between SCF and U imply that U is the cause for high SCFs in the mid- and high latitudes.

3. **Results, Section 3.1, Lines 384-385:** Again, another sentence insinuating that correlation implies causation.

4. **Results, Section 3.2, Lines 396-398 and 405-408:** These are yet more statements that read as though correlation implies causation. One cannot conclude from these correlations that the meteorological parameters examined in the study impact the SCF, but only that these results provide further evidence to support previous studies that have established a causal effect. Too many of such statements occur in the manuscript to list. Please revise the manuscript bearing in mind that correlation does not imply causation and eliminate all statements that imply a causal effect strictly from only a correlation analysis.

5. **Dataset and methods, Section 2.1:** Please provide more details on the GOCCP-CALIPSO product. How exactly is this product more consistent with the CALIPSO simulator within COSP (line 172: "fully-consistent" is too general)? Technical details starting with how the Level 1 CALIPSO data is processed is recommended. How are horizontally-oriented ice particles treated in the product? Were daytime or nighttime data used? Were there quality checks to minimize misclassification of thin cloud layers with aerosols?

6. **Results, Section 3.2, Lines 396-406:** The mechanism of how horizontally-oriented crystals are eliminated by strong horizontal winds is not clearly explained. Please clarify. Next, as mentioned in the previous comment, does your data include horizontally-oriented particles in the first place? And if so, did you look into whether the frequency of occurrence of these particles actually decreased before making this conclusion? Please clarify.

7. **Results, Section 3.2, Lines 362-370:** Please provide a clear physical explanation for why positive (negative) correlations exist between LTSS and SCF over land (ocean) in the mid- and high latitudes.

8. **Dataset and Methods, Section 2.3:** Why wasn't the Level 3 product used? The Level 3 product was further processed and screened to remove some misclassifications between thin clouds and aerosols. The screening affected mostly cirrus clouds, but it may be worth repeating the analysis with this product to check whether the results are consistent.

9. **Results, Section 3.3, Fig. 11:** Have the authors tried grouping the high, medium and low RAFs into bins with different thresholds? Would the results be robust to having bins with different thresholds? I recommend the authors to redo the analysis using different threshold values for their high, medium and low RAF bins to check how robust their results are to the choice of RAF bins.

   The extratropics of both the northern and southern hemispheres are grouped into a single category even though the aerosol loadings are very different for these two regions. I suggest that the authors separate their analysis of the extratropics into the northern and southern hemispheres and/or over ocean and land. This may help clarify the contributions from the various regions and enable more generalizations of the statistical patterns seen. For example, this could help to better interpret the "U"-shaped pattern seen in Fig. 11d.

**3 Minor Comments**

1. **Abstract, Line 40:** Please specify the months of the year. There were some changes to CALIPSO prior to November 2007.

2. **Abstract, Line 55:** Insert "more" after "relatively".

3. **Abstract, Line 60:** There is a word missing in between "regional" and "of". Perhaps "influence"?

4. **Introduction, lines 72-74:** It is misleading to discuss cirrus clouds in this context. The authors are referring to a layer of mixed-phase clouds at the same isotherm, not two different layers of clouds at two different altitudes and temperatures. The difference in the optical thickness of the mixed-phase clouds due to the liquid and ice partitioning is what is important, and not the greenhouse effect. The authors may also mention the difference in the lifetime effect depending on the liquid and ice partitioning. There should also be a reference here.

5. **Introduction, lines 84-85:** It would be clearer if the term "mixed-phase cloud" was mentioned here. The terminology is used later, but never defined upfront. Here would be a good place to introduce it.

6. **Dataset and methods, line 185:** Mlmenstadt $\rightarrow$ Mulmenstadt

7. **Dataset and methods, line 201-202:** Please provide a reference here.

8. **Results, Section 3.1, line 312:** $30 \rightarrow -30$

9. **Results, Section 3.2, Lines 419-421:** Have the authors checked that the high RAFs here are indeed due to dust and polluted dust? It may seem intuitive to make such an assumption, but the RAFs also include

smoke and it is not clear that it does not dominate the RAFs rather than dust and polluted dust without verification.

---

## Referee Report (RR2)

The overall quality of the manuscript has improved and I appreciate the efforts of the authors in taking all of my suggestions into account. I still have a few more suggestions. Please see the comments below.

- Please have a native speaker proofread the manuscript. There are too many grammatically incorrect sentences present in the current version.
- Lines 70-73: There needs to be a reference after this statement, e.g.
1) Tan, I., T. Storelvmo, and M. D. Zelinka. "Observational constraints on mixed-phase clouds imply higher climate sensitivity." *Science* 352.6282 (2016): 224-227.
2) McCoy, D. T., Hartmann, D. L., Zelinka, M. D., Ceppi, P., and Grosvenor, D. P. (2015). Mixed-phase cloud physics and Southern Ocean cloud feedback in climate models. *Journal of Geophysical Research: Atmospheres*, *120*(18), 9539-9554.
3) Tsushima, Yoko, S. Emori, T. Ogura, M. Kimoto, M. J. Webb, K. D. Williams, M. A. Ringer, B. J. Soden, B. Li, and N. Andronova. "Importance of the mixed-phase cloud distribution in the control climate for assessing the response of clouds to carbon dioxide increase: a multi-model study." *Climate Dynamics* 27, no. 2-3 (2006): 113-126.
- Lines 116-118: I know what the authors are saying here, but this needs to be more thoroughly explained to a reader who is not familiar with this study.
- Section 3.2: I appreciate the explanations provided by the authors, but this section is not well-organized and needs to be re-written for the sake of the reader.
- Figure 11 and lines 547-556: Coming back to this, the original results combining the two hemispheres shown in the second round of revisions (originally d to f) should be shown here instead of the results separating the southern hemisphere (new figures g to i), the reason being that the southern hemisphere has far fewer aerosols compared to the northern hemisphere. Thus, just as how the correlations weaken or even vanish at colder temperatures as the authors have shown, the correlations between SCFs and aerosol frequencies are less likely to be statistically significant in the southern hemisphere, as the authors have already pointed out on lines 552-553 (the confidence level was reduced). It would therefore be more appropriate to show the more statistically robust results shown in the original Figure 11 instead of the less statistically robust results presented in the current version of the manuscript. Moreover, the fact that the aerosol product used in this study was the Level 2 product, which does not have the additional level of screening that the Level 3 product that Tan et al. (2014) used, adds to the level of uncertainty.
- Lines 560- 564: This statement implies a causal relationship based on statistical correlations, which is incorrect. One cannot be "certain" from the analysis that the meteorological parameters examined in the study impact the variation of SCFs. The authors need to be much more careful in their language and avoid making causal statements.

---

## Referee Report (RR3)

**Review of the paper "Effects of atmospheric dynamics and aerosols on the thermodynamic phase of cold clouds" by Jiming Li et al.**

I appreciate that the authors rewrite their introduction to get rid of plagiarism and put effort to even improve it. They also better describe the observations and the method used to produce the SLFs. However, I still have major concerns about the manuscript - developed below -, which should be treated before considering publication.

**Validity of the results**

I found their results not consistent with previously published material using CALIPSO-GOCCP. In Cesana et al. (2015) Fig. 5 bottom right, the ratio of ice to liquid+ice (i.e., 1-SLF) is shown as a function of the temperature and the latitude. At -10˚C it is quite constant regardless of the latitude and the value ranges between 60 and 80%. In Cesana and Chepfer (2013) Fig. 7b, the global average of the ratio of ice to ice+liq is represented as a function of the temperature for different latitudinal bands and again, for -10˚C, the SLF is higher than 70% for all regions.

I could not reproduce the author's results although I used their formula and the CALIPSO-GOCCP monthly data. I finally figured out that was because they used day + night time data. As mentioned in Cesana and Chepfer (2013), Cesana et al (2015,2016), daytime data are nosier because of solar contamination – which particularly affect the perpendicular laser channel – and should not be used for statistical analysis. It affects even more overly bright regions such as stratocumulus regions that reflect a lot of incoming solar light (even more in the tropics).

I enclosed below my results using the authors' formula SLF = cltemp_liq / (cltemp_liq + cltemp_ice) for night and daytime CALIPS-GOCCP 2007-2015 monthly data, on separate figures.

[Figure]

Figure 1: Seasonal variation of the Cloud Phase Ratio (liq/(liq+ice) for different isoterm (-10, -20 and -30°C, from the left to the right) using daytime GOCCP monthly data (2007-2015)

[Figure]

**Figure 2: Seasonal variation of the Cloud Phase Ratio (liq/(liq+ice) for different isoterm (-10, -20 and -30°C, from the left to the right) using nighttime GOCCP monthly data (2007-2015)**

As you can see, results are quite different, especially over subsidence regime regions. I suggest the authors start over again using nighttime only data.

As the rest of the study is based on the calculation of the SLF, it completely questions the validity of the following results; this could explain why the authors found that higher large-scale vertical velocity and relative humidity promote the fraction of supercooled liquid at very low temperatures, which is in disagreement with Cesana et al (2015) results, who used the same CALIPSO-GOCCP and ERAi data.

**Large-scale vs. in-cloud**
There is a confusion between large-scale and in-cloud meteorological parameters.
Large-scale velocity gives an information about the gridbox averaged vertical velocity and thus the type of cloud regime to expect. Yet it does not mean the in cloud vertical velocity is necessarily very large and the authors also reference papers that used in-cloud updrafts velocity rather than large-scale vertical velocity without mentioning it. They should clear make the distinction in the manuscript.

**Introduction**

While the authors substantially re-wrote the introduction – and it is a good thing-, they still don't really explain why they want to focus on the relation aerosol – phase other than it wasn't done before. They could reduce it by skipping most of the second paragraph for example - why do you focus on water vapor and size and shape of ice crystal whereas you don't investigate this at all in your study? – and add more detail about why they want to focus on aerosol – phase relation.

**Minor comments**

Line 79 and 84: the use of the word important is not appropriate here. Please consider reformulating.

Line 92 92: Redundant use of also and addition/additionally

Line 105: dominates?

Line 112 – 116: This is confused: content of ice in ice clouds? Discrepant?
Line 126 I would rather say detailed than accurate.

Line 126: Define the CALIPSO acronym

Line 136 – 141: This is not correct. Cesana et al. (2015) analyzed what you called here cloud phase changes (the cloud phase transition) at global scale in obs and models.
Similar comment for line 468

Line 139 What is systematic studies?

Line 171: for single scattering only. Otherwise liquid droplets also produce cross polarization - because of multiple scattering issues – but relatively less than ice crystals. "Spherical particles typically do not"

Line 181 differences **between** the observations … outputs are **mostly** attributed . Indeed, Line 182 Cesana et al., 2015 do not use the lidar simulator. You might reference Cesana and Chepfer 2012 instead and add e.g. because they are many other papers out there that use the lidar simulator.

Line 192 (e.g. Chepfer et al., 2013)
Line 194 further classifies

Line 224 231: The impact of the oriented ice crystals on the mixed phase cloud retrievals is negligible after the lidar tilt (late 2007) [Cesana et al., 2016].

Line 301: "are" should not be here.

Line 325 334: Do you mean anti-correlated? So you expect to have fewer SCF in regions of large values of aerosol such as the mid-lat? Yet mid-lat regions are known to known to be the regions where mixed-phase clouds form.

Line 417: Seasonal variations at high latitude imply that only daytime or nighttime data were used, which might explain why different correlations are found at mid-lat (using both day and nighttime data no matter the season) and high lat.

Line 430: Again, Cesana et al (2015) mentioned the large-scale vertical velocity whereas West et al (2014) the subgrid, which is different. Although an increase of the LWP does not say anything about the mixed-phase clouds.

The rest of the paper is very difficult to interpret as the variation of SCF cannot be trusted.
I won't comment the results section as I believe the results are incorrect.

---

## Author Response (AR2)

**Response to Reviewer #1's Comments:**

**Jiming Li et al. (Author)**

**We are very grateful for the Review #1's detailed comments and suggestions, which help us improve this paper significantly. Based on two Reviewers' comments, we made a lot of important changing in the revised paper. The detailed information includes:**

(1) Some grammatical errors and unreasonable presentations already were corrected in the revision and the paper also be edited by the nature language editing service to make it more readable.

(2) We reorganized the introduction section. In addition, some superfluous information in each section was deleted and some interpretations in each section were added in order to make the manuscript more clear. In particular, we rechecked the results of Fig.1, 2 and 3, and added the detailed information about the CALIPSO-GOCCP product (such as, technical details) and method to calculate the SCF in the section 2.1.

(3) In the section 3 (results part), we also added some physical interpretations of the correlations.

**Major Comments:**

We appreciated the insightful suggestion and comments made by reviewers. Some responses to the reviewer's comments are listed below:

1. Results, Section 3.1, Lines 354-356: The authors write that the negative correlations between skin temperature and SCF in the mid- and high latitudes "mean" that high skin temperature promotes the glaciation of supercooled droplets, and that it the positive correlations "mean" that high skin temperature inhibits glaciation in the tropics. Firstly, correlation does not imply causation. These correlations may be caused by many different confounding factors that lead to mechanisms that are completely unrelated to skin temperature. What is the physical mechanism for why high skin temperature promotes glaciation in the mid- and high latitudes? This was never mentioned in the manuscript.

The authors explain that high skin temperature in the tropics inhibits glaciation because it triggers deep convection, which lofts the liquid droplets to the colder isotherms, but what is the mechanism for the mid- and high latitudes? Secondly, the area of the tropics that the authors are referring to appears to be only a small part of the entire tropics (how many grid boxes/what is the percent- age of the area of the tropics that is positively correlated?). Thirdly, if high skin temperature is indeed inhibiting glaciation due to a triggering of deep convection in the tropics, then shouldn't the coldest isotherm, i.e. $-30 \circ$ C also show positive correlations in the tropics as well? The correlations between relative humidity and SCF and vertical velocity and SCF appear to support your argument in the tropics, but please clarify the effect of skin temperature.

**Response:** We appreciated the insightful suggestion and comments made by reviewer #1. Indeed, the changes of cloud phase composition include many confounding factors, e.g., ice nuclei (IN) or dynamical processes. We cannot provide certain causation only through the correlation analysis. In this paper, our main purpose is to indicate the aerosols' effect on nucleation cannot fully explain the regional variation and seasonal cycles of supercooled liquid cloud fraction. Although these statistical correlations can't imply certain causation, we expect that these results may provide a unique point of view on the phase change of mixed-phase cloud. For the questions from Reviewer #1, we already added some physical interpretations of the correlations in the section 3.1.

**Question:**

Next, there is an apparent contradiction between the authors' explanation of the correlations found in the tropics and Figures 1, 2 and 3. The authors explain that vigorous convective activity, high relative humidity as well as high vertical velocities in this region cause supercooled liquid to loft to higher altitudes too quickly to allow for glaciation of supercooled liquid, which would suggest that SCFs should be higher in this region for the aforementioned reasons. Yet, Figs. 1, 2 and 3 all show that the tropics contain mixed-phase clouds with some of the lowest SCFs in the world. Please clarify.

**Response:** Yes, Fig1, 2 and 3 all show that the tropics have the lowest SCFs in the world at any isotherm. This result is consistent with previous study from Hu et al.(2010). We consider the possible reasons as: (1) The CALIOP can't penetrate the optically thick

clouds (optical depth>3) to detect the lower clouds, especially over the tropics where cloud overlap is very frequent. (2) This study focuses on the cloud top phase of not only mixed-phase cloud, but also those pure ice or liquid clouds. It means that frequent tropical ice cloud may results in low SCF. (3) Those regions with lowest SCF usually correspond to the typical descending motion. In these moderate or strong subsidence regions, the cloud occurrence is infrequent at the atmospheric layer between -40 ℃ to 0 ℃. However, a little high cloud (or ice phase cloud) still can exist at this layer. The major source of these high clouds is topography-driven gravity wave activity, advection from neighboring tropical convection centers such as the Amazon Basin, the Congo Basin, or ascent associated with mid-latitude fronts (Yuan and Oreopoulos, 2013). Thus, positive correlation between SCF and velocity also exists at subsidence regions with low SCF.

2. Abstract, Lines 57-59: This is another example of a sentence that reads as if correlations between SCF and U imply that U is the cause for high SCFs in the mid- and high latitudes.
**Response:** We already revised it.

3. Results, Section 3.1, Lines 384-385: Again, another sentence insinuating that correlation implies causation.
**Response:** We already revised it.

5. Dataset and methods, Section 2.1: Please provide more details on the GOCCP-CALIPSO product. How exactly is this product more consistent with the CALIPSO simulator within COSP (line 172: "fully-consistent" is too general)? Technical details starting with how the Level 1 CALIPSO data is processed is recommended. How are horizontally-oriented ice particles treated in the product? Were daytime or nighttime data used? Were there quality checks to minimize misclassification of thin cloud layers with aerosols?
**Response:** In the revised paper, we already added the detailed information about the CALIPSO-GOCCP product (such as, technical details) and method to calculate the SCF. For above questions, some interpretations are added in the section 2.1 in order to make

the manuscript more clear.

6. Results, Section 3.2, Lines 396-406: The mechanism of how horizontally-oriented crystals are eliminated by strong horizontal winds is not clearly explained. Please clarify. Next, as mentioned in the previous comment, does your data include horizontally-oriented particles in the first place? And if so, did you look into whether the frequency of occurrence of these particles actually decreased before making this conclusion? Please clarify.

**Response:** For the horizontal wind speed at 100 hPa, Noel et al. (2010) found that the frequency of oriented crystal drops severely in areas dominated by stronger horizontal wind speed at 100 hPa. This effect is especially noticeable at latitudes below 40°. But, they have not explained why the correlation between horizontal wind speed and horizontally-oriented ice particle is negative. We speculate that strong horizontal wind possible result in strong vertical wind shear, thus cause shear-gravitational wave motions to induce local updraft circulations (Rauber and Tokay, 1991)**.** As a result, updraft possible perturbs the orientation of ice crystal. In addition, Westbrook et al. (2010) pointed out that supercooled liquid water layers is very important in the formation of planar ice particles susceptible to orientation at midlatitudes (see section 2.1).

The GOCCP-CALIPSO product doesn't include the horizontally-oriented particles in their ice phase clouds, please see the section 2.1. We have to verify this conclusion in other study by using the VFM product of CALIPSO.

7. Results, Section 3.2, Lines 362-370: Please provide a clear physical explanation for why positive (negative) correlations exist between LTSS and SCF over land (ocean) in the mid- and high latitudes.

**Response:** We already added some interpretations in the section 3.2.

8. Dataset and Methods, Section 2.3: Why wasn't the Level 3 product used? The Level 3 product was further processed and screened to remove some misclassifications between thin clouds and aerosols. The screening affected mostly cirrus clouds, but it may be worth repeating the analysis with this product to check whether the results are consistent.

**Response:** We agreed with reviewer. Indeed, the Level 3 product was further improved compared with Level 2 product. However, by performing same correlation analysis with Level 2 aerosol product, we found the results are similar. As a sample, the figure below showed that the patterns of spatial correlations between skin temperature and SCF are similar. Thus, we still kept the results from Level 2 aerosol product in the revised paper.

[Figure]

9. Results, Section 3.3, Fig. 11: Have the authors tried grouping the high, medium and low RAFs into bins with different thresholds? Would the results be robust to having bins with different thresholds? I recommend the authors to redo the analysis using different threshold values for their high, medium and low RAF bins to check how robust their results are to the choice of RAF bins. The extratropics of both the northern and southern hemispheres are grouped into a single category even though the aerosol loadings are very different for these two regions. I suggest that the authors separate their analysis of the extratropics into the northern and southern hemispheres and/or over ocean and land. This may help clarify the contributions from the various regions and enable more generalizations of the statistical patterns seen. For example, this could help to better interpret the "U"-shaped pattern seen in Fig. 11d.

**Response:** Following the suggestions from Reviewer #1, we separated the extratropics into the northern and southern hemispheres to perform similar spatial analysis by using different RAFs thresholds. Related changes already were added in the section 3.3.

**Minor Comments:**

1. Abstract, Line 40: Please specify the months of the year. There were some changes to CALIPSO prior to November 2007.
**Response:** We already revised it.

2. Abstract, Line 55: Insert "more" after "relatively".
**Response:** We already added it.

3. Abstract, Line 60: There is a word missing in between "regional" and "of". Perhaps "influence"?
**Response:** We already added it.

4. Introduction, lines 72-74: It is misleading to discuss cirrus clouds in this context. The authors are referring to a layer of mixed-phase clouds at the same isotherm, not two different layers of clouds at two different altitudes and temperatures. The difference in the optical thickness of the mixed-phase clouds due to the liquid and ice partitioning is what is important, and not the greenhouse effect. The authors may also mention the difference in the lifetime effect depending on the liquid and ice partitioning. There should also be a reference here.
**Response:** We appreciated the insightful suggestions and comment made by reviewer. The introduction was reorganized in the revised paper. Some confused sentences and wrong quotations were corrected.

5. Introduction, lines 84-85: It would be clearer if the term "mixed-phase cloud" was mentioned here. The terminology is used later, but never defined upfront. Here would be a good place to introduce it.
**Response:** We already revised it.

6. Dataset and methods, line 185: Mlmenstadt → Mulmenstadt

**Response:** We already revised it.

7. Dataset and methods, line 201-202: Please provide a reference here.

**Response:** We already added the reference.

8. Results, Section 3.1, line 312: 30 → −30

**Response:** We already revised it.

9. Results, Section 3.2, Lines 419-421: Have the authors checked that the high RAFs here are indeed due to dust and polluted dust? It may seem intuitive to make such an assumption, but the RAFs also include smoke and it is not clear that it does not dominate the RAFs rather than dust and polluted dust without verification.

Response:

**Response:** Yes, we already confirm this conclusion by using Level 2 and Level 3 aerosol products. Dust and polluted dust indeed have higher RAFs than smoke in the selected grid.

Jiming Li et al. (Author)

**We are very grateful for the Review #2's detailed comments and suggestions, which help us improve this paper significantly. Based on two Reviewers' comments, we made a lot of important changing in the revised paper. The detailed information includes:**

(1) Some grammatical errors and unreasonable presentations already were corrected in the revision and the paper also be edited by the nature language editing service to make it more readable.

(2) We reorganized the introduction section. In addition, some superfluous information in each section was deleted and some interpretations in each section were added in order to make the manuscript more clear. In particular, we rechecked the results of Fig.1, 2 and 3, and added the detailed information about the CALIPSO-GOCCP product (such as, technical details) and method to calculate the SCF in the section 2.1. We hope that the added information can explain the doubt from Review #2.

(3) In the section 3 (results part), we added some physical interpretations of the correlations.

**Specific responses:**

We appreciated the insightful suggestion and comments made by reviewers. Some responses to the reviewer's comments are listed below:

(1) The method the authors used to derive the SCFs is not clear. They need to better explain how they did it. For example, there is no such -10 isotherm in the CALIPSO-GOCCP product. Did they choose one 3deg-temperature bin only for each -10 -20 -30 isotherm? I personally couldn't reproduce their figure 1 2 and 3 using CALIPSO-GOCCP data. Anyhow, I found particularly difficult to believe there are almost no SLF over shallow cumulus region around –10degC, which is why I doubt the results presented here. Even if the calculations were correct, the result section is far too

descriptive and there are no discussions behind the description. The authors should revisit their results sections to make it shorter more clear and delete all descriptions irrelevant to the discussion.

**Response:** We are very sorry to mislead reviewers about Fig.1, 2 and 3. We rechecked the results of Fig.1, 2 and 3, and added the detailed information about the CALIPSO-GOCCP product (such as, technical details) and method to calculate the SCF in the section 2.1. In section 3, we also added some physical interpretations of the correlations to make the manuscript more clear.

(2) I don't understand the first paragraph of section 3. The authors seem to pick randomly parameters and try to find a correlation between them. It is very confusing and need to be fully re-written. Again, the authors have to pay attention to the explanations and clearly state why they chose one parameter over another and why did they change with respect to the bands of latitude.

**Response:** We appreciated the insightful suggestion from reviewer #2. In the revised paper, we reorganized the section3 and added some physical interpretations. Based on the spatial distributions of correlations, we choose different meteorological parameters to analyze the spatial correlations in different regions. For example, the correlations (at the 90% confidence level) between SCF and velocity (or u wind) almost locate at tropics (or middle latitudes).

(3) In general the method to compute the figures is unclear and blurry. For example, how did they get this 10x10 grid from the 2x2 in the correlation figures. What about the results in Fig.11.

**Response:** In the revised paper (section 3.2), we added the description of method. In addition, we also changed the grid size from 10x10 to 6x6. For the Fig. 11, the results are based on the spatial correlation analysis for those $6º \times 6º$ grids, which have correlation with high confidence level (90% level).

(4) Line 69 This sentence is wrong. The IPCC statement is as follow: "Cloud feedbacks remain the largest source of uncertainty in climate models" ipcc 2007/2013

**Response:** We already revised it.

(5) Line 97-103 The second part (However...) is confusing, please reformulate by adding something like may coexist "for a certain time" as opposed to "in equilibrium" found in the first part.

**Response:** We already reformulated it.

(6) Line 184 It is possible that GOCCP slightly underestimates the ice clouds in the lowest layers in the Arctic but not in the mid-lat cases as demonstrated in Cesana et al ., 2016 (Cesana G.,H. Chepfer, D. Winker, X. Cai, B. Getzewich, H. Okamoto, Y. Hagihara, O. Jourdan, G. Mioche,V. Noel and M. Reverdy, 2016: Using in-situ airborne measurements to evaluate three cloud phase products derived from CALIPSO, J. Geophys. Res. Atmos., 121, doi:10.1002/2015JD024334).

**Response:** We agreed with reviewer and already revised it in the revised paper.

(7) Line 188: which version? Day / night / day+night?

**Response:** Please see the section 2.1.

(8) Line 202 Does that mean you use the bin -9 to -12 for -10degC? Please clarify

**Response:** We added the related information in the section 2.1.

(9) Line 271: 10x10 seems very large. Did you try a sensitivity test at 5x5 that should be enough to gather statistically representative samples over 8 years.

**Response:** We already changed the grid size from $10°\times10°$ to $6°\times6°$.

---

## Author Response (AR3)

**Response to Reviewer #1's Comments:**

**Jiming Li et al. (Author)**

**We are very grateful for the Review #1's detailed comments and suggestions, which help us to improve this paper significantly. We made some revisions based on two reviewers' comments. The detailed information includes:**

(1) Some grammatical errors and inaccurate statements already were corrected in the revision and the paper also be edited by the nature language editing service to make it more readable.

(2) Based on the reviewer #2's suggestion, we restart this investigation using only nighttime data to avoid artifacts due to noise from scattering of sunlight, thus some statistical results are different from original version. But, it is noting that different result does not mean that original conclusions are wrong.

As stated by reviewer #2, the inconsistency is because that our results are based on the day+night time data, whereas the studies from Cesana et al. (2015; 2016) only used the nighttime cloud phase. Indeed, it is well known that strong solar noise can contaminate the lidar signal during daytime and cause the uncertainty of cloud phase product. In the previous versions of our paper (Line 285-295), we also emphasized that "to avoid artifacts due to noise from scattering of sunlight, it is better to conduct the CALIOP retrieval during nighttime. However, in view of the lack of CALIPSO observations at high latitudes of the northern Hemisphere during boreal summer nights, this study utilizes the mean values of SCFs, meteorological parameters and RAFs during daytime and nighttime to perform the temporal and spatial correlations analysis". However, Sassen et al. (2008) have pointed out that the effect of CALIOP signal noise from scattered sunlight only can cause a small part uncertainty of the observed day–night variations in cirrus. The diurnal cirrus patterns mostly still reflect real cloud processes (Sassen et al., 2009). Based on their conclusions, we possible infer that the obvious different patterns of SCF during day- and night-time can't be fully attributed to solar noise signature. However, to minimize the impact of SCF during daytime due to solar noise signature on the statistical results, we followed the suggestion from reviewer#2 to perform same analysis by using the nighttime only data in the revised paper.

**Special responses:**

**(1) Lines 70-73: There needs to be a reference after this statement, e.g…**

**Response:** we already added the related references after this statement.

**(2)Lines 116-118: I know what the authors are saying here, but this needs to be more thoroughly explained to a reader who is not familiar with this study.**

**Response:** We already re-organized this sentence and make it more readable.

**(3) Section 3.2: I appreciate the explanations provided by the authors, but this section is not well-organized and needs to be re-written for the sake of the read.**

**Response:** We already revised this section in the revised paper.

(4) Figure 11 and lines 547-556: Coming back to this, the original results combining the two hemispheres shown in the second round of revisions (originally d to f) should be shown here instead of the results separating the southern hemisphere (new figures g to i), the reason being that the southern hemisphere has far fewer aerosols compared to the northern hemisphere. Thus, just as how the correlations weaken or even vanish at colder temperatures as the authors have shown, the correlations between SCFs and aerosol frequencies are less likely to be statistically significant in the southern hemisphere, as the authors have already pointed out on lines 552-553 (the confidence level was reduced). It would therefore be more appropriate to show the more statistically robust results shown in the original Figure 11 instead of the less statistically robust results presented in the current version of the manuscript. Moreover, the fact that the aerosol product used in this study was the Level 2 product, which does not have the additional level of screening that the Level 3 product that Tan et al. (2014) used, adds to the level of uncertainty.

**Response:** We appreciated the insightful suggestion from reviewer #1. Indeed, the southern hemisphere and tropics have far fewer aerosols compared to the northern hemisphere. Thus, the correlations between SCFs and aerosol frequencies are less likely to be statistically significant in the southern hemisphere and tropics. In the revised paper, we only presented the global results by combining the two hemispheres. The statistical results show that the impact of aerosol on the SCFs is obvious at a global scale and a fixed isotherm (such as, -20 ℃). That is, the SCFs almost decrease with increasing RAF (please section 3.3). In addition, we also agree the comment from reviewer #1: "the aerosol product used in this study was the Level 2 product, which does not have the additional level of screening that the Level 3 product that Tan et al. (2014) used, adds to the level of uncertainty". But, by performing same correlation analysis with Level 3 aerosol product in the second round of revisions, we found the results are similar.

**Response to Reviewer #2's Comments:**

**Jiming Li et al. (Author)**

**We are very grateful for the Review #2's detailed comments and suggestions, which help us to improve this paper significantly. Meantime, the reviewer#2 also did lots of laudable efforts to reproduce our results, and help us to interpret why our results are different from previous studies (such as, Cesana et al. (2015)). Our responses are below:**

(1) Some grammatical errors already were corrected in the revision and the paper also be edited by the nature language editing service to make it more readable.

(2) Based on the reviewer #2's suggestion, we restart this investigation using only nighttime data to avoid artifacts due to noise from scattering of sunlight, thus some statistical results are different from original version.

As stated by reviewer #2, the inconsistency is because that our results are based on the day+night time data, whereas the studies from Cesana et al. (2015; 2016) only used the nighttime cloud phase. Indeed, it is well known that strong solar noise can contaminate the lidar signal during daytime and cause the uncertainty of cloud phase product. In the previous versions of our paper (Line 285-295), we also emphasized that "to avoid artifacts due to noise from scattering of sunlight, it is better to conduct the CALIOP retrieval during nighttime. However, in view of the lack of CALIPSO observations at high latitudes of the northern Hemisphere during boreal summer nights, this study utilizes the mean values of SCFs, meteorological parameters and RAFs during daytime and nighttime to perform the temporal and spatial correlations analysis". However, Sassen et al. (2008) have pointed out that the effect of CALIOP signal noise from scattered sunlight only can cause a small part uncertainty of the observed day–night variations in cirrus. The diurnal cirrus patterns mostly still reflect real cloud processes (Sassen et al., 2009). Based on their conclusions, we possible infer that the obvious different patterns of SCF during day- and night-time can't be fully attributed to solar noise signature. However, to minimize the impact of SCF during daytime due to solar noise signature on the statistical results, we followed the suggestion from reviewer#2 to perform same analysis by using the nighttime only data in the revised paper.

**Special responses:**

**(1) Large-scale vs. in-cloud**

There is a confusion between large-scale and in-cloud meteorological parameters. Large-scale velocity gives an information about the grid box averaged vertical velocity and thus the type of cloud regime to expect. Yet it does not mean the in cloud vertical velocity is necessarily very large and the authors also reference papers that used in-cloud updrafts velocity rather than large-scale vertical velocity without mentioning it. They should clear make the distinction in the manuscript.

**Response:** We agree with the comment from the reviewer #2. Indeed, the large-scale velocity is different from the in-cloud updrafts velocity, we already made the explanation in the section 2.2 of the revised paper.

**(2) Introduction :**

While the authors substantially re-wrote the introduction – and it is a good thing-, they still don't really explain why they want to focus on the relation aerosol – phase other than it wasn't done before. They could reduce it by skipping most of the second paragraph for example - why do you focus on water vapor and size and shape of ice crystal whereas you don't investigate this at all in your study? – and add more detail about why they want to focus on aerosol – phase relation.

**Response:** We already re-organized the second paragraph in the revised paper.

**(3) Line 112-116: This is confused: content of ice in ice clouds? Discrepant?**

**Response:** We already revised it.

**(4) Line 171: for single scattering only. Otherwise liquid droplets also produce cross polarization - because of multiple scattering issues – but relatively less than ice crystals. "Spherical particles typically do not".**

**Response:** We agree with reviewer. It was already revised.

Some responses to other minor comments also were added in the revised paper.

[revised manuscript text omitted]

---

## Author Response (AR4)

**Response to Reviewer #1's Comments:**

**Jiming Li et al. (Author)**

We are very grateful for the Review #1's final comments and suggestions, which help us to improve this paper significantly. Based on the reviewer' comments, we already made some revisions and implemented the final edits suggested by the reviewer.